# Additional feedforward mechanism of Parkin activation via binding of phospho-UBL and RING0 in *trans*

Dipti Ranjan Lenka[1], Shakti Virendra Dahe[1], Odetta Antico[2], Pritiranjan Sahoo[1], Alan R Prescott[3], Miratul MK Muqit[2], Atul Kumar[1]*

[1]Department of Biological Sciences, Indian Institute of Science Education and Research (IISER) Bhopal, Bhopal, India; [2]MRC Protein Phosphorylation and Ubiquitylation Unit, School of Life Sciences, University of Dundee, Dundee, United Kingdom; [3]Division of Cell Signalling and Immunology, Dundee Imaging Facility, School of Life Sciences, University of Dundee, Dundee, United Kingdom

**Abstract** Loss-of-function Parkin mutations lead to early-onset of Parkinson's disease. Parkin is an auto-inhibited ubiquitin E3 ligase activated by dual phosphorylation of its ubiquitin-like (Ubl) domain and ubiquitin by the PINK1 kinase. Herein, we demonstrate a competitive binding of the phospho-Ubl and RING2 domains towards the RING0 domain, which regulates Parkin activity. We show that phosphorylated Parkin can complex with native Parkin, leading to the activation of auto-inhibited native Parkin in *trans*. Furthermore, we show that the activator element (ACT) of Parkin is required to maintain the enzyme kinetics, and the removal of ACT slows the enzyme catalysis. We also demonstrate that ACT can activate Parkin in *trans* but less efficiently than when present in the *cis* molecule. Furthermore, the crystal structure reveals a donor ubiquitin binding pocket in the linker connecting REP and RING2, which plays a crucial role in Parkin activity.

*For correspondence:
atul@iiserb.ac.in

## eLife assessment

This is a **useful** manuscript describing the competitive binding between Parkin domains to define the importance of dimerization in the mechanism of Parkin regulation and catalytic activity. The evidence supporting the importance of Parkin dimerization for an 'in trans' model of Parkin activity described in this manuscript is **solid**, but lacks more stringent and biochemical characterization of competitive binding that could provide more direct evidence to support the author's conclusions. This work will be of interest to those focused on defining the molecular mechanisms involved in ubiquitin ligase interactions, PINK-Parkin-mediated mitophagy, and mitochondrial organellar quality control.

## Introduction

Parkinson's disease (PD) is a neurodegenerative disorder characterized by the loss of dopaminergic neurons in the substantia nigra leading to motor defects. PD is primarily sporadic, occurring mainly in older people. Mutations in several genes, such as *PARK2* (Parkin) and *PARK6* (PINK1, PTEN-induced kinase 1), cause early-onset autosomal recessive juvenile parkinsonism (ARJP). Parkin and PINK1 function together in a common mitochondrial homeostasis pathway in which damaged mitochondria are cleared by autophagy (mitophagy; *Bonifati et al., 2002*; *Martin et al., 2011*; *Kitada et al., 1998*; *Valente et al., 2004*; *Chung et al., 2016*; *Exner et al., 2012*; *Narendra et al., 2012*).

Parkin is an autoinhibited RBR family E3 ubiquitin ligase (*Chaugule et al., 2011*) consisting of an N-terminal ubiquitin-like (Ubl) domain followed by four $Zn^{2+}$ binding domains RING0, RING1, in-between-RING (IBR), and RING2 (*Spratt et al., 2014*). Parkin is a cytosolic protein activated following mitochondrial stress, mediated by PINK1 phosphorylation of Serine 65 (S65) on ubiquitin. Phosphorylation of ubiquitin enhances binding with Parkin and leads to the recruitment of Parkin to sites of damaged mitochondria (*Kane et al., 2014*; *Kazlauskaite et al., 2014*; *Koyano et al., 2014*). On mitochondria, S65 of the Ubl domain of Parkin is phosphorylated by PINK1 (*Kazlauskaite et al., 2014*; *Kondapalli et al., 2012*; *Shiba-Fukushima et al., 2014*; *Shiba-Fukushima et al., 2012*), resulting in a fully active Parkin conformation. Fully active Parkin attaches new ubiquitin molecules on mitochondrial proteins, which are phosphorylated by PINK1 to recruit more cytoplasmic Parkin to the mitochondria, thus resulting in a positive feedforward amplification cycle (*Ordureau et al., 2014*). Ubiquitination of mitochondrial proteins by Parkin also leads to the recruitment of autophagy receptors required for mitophagy (*Tanaka et al., 2010*; *Chan et al., 2011*).

Like other RBR-family E3 ligases, Parkin binds to an E2, and ubiquitin is transferred from E2 onto the catalytic C431 residue (on RING2) of Parkin before ubiquitination of lysines on target substrates (*Wenzel et al., 2011*; *Walden and Rittinger, 2018*). On Parkin, several elements are present that maintain autoinhibited conformation of Parkin. The E2 binding site on RING1 is blocked by the Ubl domain and the short repressor (REP) element. Furthermore, C431 on RING2 is occluded by the RING0 domain of Parkin, which inhibits Parkin activity (*Chaugule et al., 2011*; *Riley et al., 2013*; *Trempe et al., 2013*; *Wauer et al., 2015*). The phospho-Ubl domain binds within a basic patch (comprising K161, R163, and K211) on RING0 and displaces RING2 to expose C431 to activate Parkin (*Gladkova et al., 2018*). In the structure of phospho-Parkin with RING2 removed, an activating element (ACT, 101–109), which is present in the linker region (77-140) between Ubl and RING0 domains, binds on the RING0 interface (*Gladkova et al., 2018*). Mutations in the ACT are shown to affect Parkin activity negatively (*Gladkova et al., 2018*), suggesting their importance in Parkin regulation. Phospho-ubiquitin (pUb) binds in a pocket between RING0 and RING1, and activates Parkin allosterically (*Wauer et al., 2015*; *Sauvé et al., 2015*; *Kumar et al., 2015*). pUb binding results in the displacement of the IBR domain, and the straightening of helix-1 of the RING1 domain (*Kumar et al., 2017*). Massive domain rearrangements have been proposed in the active state to allow the transfer of donor ubiquitin (bound between helix-1 and IBR) from E2 (on RING1) to C431 (on RING2) of Parkin (*Gladkova et al., 2018*; *Kumar et al., 2017*; *Sauvé et al., 2018*; *Condos et al., 2018*).

Several crystal structures of Parkin were solved in the last decade using various truncations in Parkin, which revealed new insights into the conformational changes during the intricate activation process of Parkin (*Figure 1—figure supplement 1A*). A few years ago, using the structure of truncated phospho-Parkin (RING2 removed; *Figure 1—figure supplement 1A*), a model of phospho-Parkin was proposed wherein RING2 would be displaced from RING0 to occupy a pocket near the IBR domain (*Figure 1—figure supplement 1B*; *Gladkova et al., 2018*; *Sauvé et al., 2018*). However, the extent of conformational changes and domain rearrangements due to different regulatory elements of Parkin in the active state remains elusive. For example, it is not clear how and by what mechanism the displaced pUbl from RING1 would be recognized on RING0 in the *cis* molecule (as per the proposed model in *Figure 1—figure supplement 1B*) and not in the *trans* molecule, especially considering the likelihood of an encounter with a *trans* molecule in the crowded molecular environment. Previous cellular data co-expressing WT-Parkin and mutant Parkin constructs suggested the self-association of Parkin molecules after PINK1 activation at sites of damaged mitochondria (*Lazarou et al., 2013*). However, a role for phospho-ubiquitin-mediated recruitment of mutant Parkin, induced by co-expressed wild-type Parkin, could not be excluded. Furthermore, structural studies to understand the Parkin activation mechanism in the last decade have not captured any dimerization of Parkin in vitro (*Wenzel et al., 2011*; *Walden and Rittinger, 2018*; *Riley et al., 2013*; *Trempe et al., 2013*; *Wauer and Komander, 2013*; *Gladkova et al., 2018*; *Wauer et al., 2015*; *Kumar et al., 2015*; *Sauvé et al., 2015*; *Kumar et al., 2017*; *Sauvé et al., 2018*; *Condos et al., 2018*).

Herein, using X-ray crystal structures, biophysical methods, and in vitro assays, we demonstrate the *trans* conformational changes in Parkin during the activation process, revealing novel insights into the Parkin activation mechanism. Our data suggest that the phospho-Ubl (pUbl) domain transiently binds to the basic patch on RING0 and competes with the RING2 domain. In addition to the previous observation that pUbl binding results in RING2 displacement, our new data show that the presence

of RING2 restricts the binding of pUbl with the Parkin core, which establishes the competitive mode of interaction between RING2 and pUbl. The crystal structure of pUbl-linker (1-140) depleted Parkin (141-465)-pUb complex and supporting data show that RING2 is displaced transiently during the activation process and returns to its closed state after the removal of the pUbl domain from phospho-Parkin, suggesting dynamic nature of conformational changes during Parkin activation. Furthermore, we report Parkin dimerization, mediated by interactions between pUbl and the basic patch on the RING0 domain in *trans*. We also demonstrate that phospho-Parkin activates autoinhibited Parkin in *trans*, suggesting an additional feedforward mechanism of Parkin activation. Our data also reveals new insights into the regulation mediated by the ACT of Parkin, wherein the ACT is required for maintaining the enzyme kinetics. We show that similar to phospho-Ubl, ACT can also work in *trans*, although ACT is more efficient in *cis*. Furthermore, using X-ray crystallography and supporting experiments, we have characterized a new donor ubiquitin binding site in the linker region (408-415) between the REP element and RING2, which plays a crucial role in Parkin activity.

## Results

### Incorporation of molecular scissors to capture intricate dynamic conformations on Parkin

Previous studies using various biophysical methods showed that upon phosphorylation of the Ubl domain of Parkin, phospho-Ubl (pUbl) does not interact with the core of Parkin, lacking the Ubl domain (*Wauer et al., 2015*; *Kumar et al., 2015*; *Sauvé et al., 2015*). However, the crystal structure of phospho-Parkin missing the RING2 (1-382) showed pUbl domain bound to the basic patch (K161, R163, K211) on the RING0 domain (*Gladkova et al., 2018*; *Sauvé et al., 2018*). RING2 shared a large surface with RING0, and the superimposition of phospho-Parkin (1–382, PDBID: 6GLC) and WT-Parkin (PDBID: 5C1Z) structures showed steric clashes between RING2, ACT, and pUbl (*Figure 1A*). Therefore, we hypothesized whether the RING2 domain competes with the pUbl domain and thus blocks the interaction of pUbl with RING0. The latter hypothesis would also explain why previous attempts to study pUbl interactions show weak or no interactions between pUbl and Parkin in *trans*.

To capture crystal structures of protein-protein complexes, researchers use fusion constructs to allow the expression of two proteins in a single polypeptide chain. The fusion method increases the effective net concentration of two proteins in solution compared to mixing two proteins separately, thus stabilizing the interactions between two proteins. Earlier binding assays on Parkin failed to capture interactions in *trans*, and we speculated that this might be due to the lower net concentration of the domain in *trans* compared to the high net concentration of the fused domain. We hypothesized that untethering (cleavage of peptide bond) upon protease treatment would solve the above problem and enable us to capture the binding in *trans* using biophysical methods. To understand the above intricate mechanism, we introduced molecular scissors human rhinovirus type 3C (HRV 3C) protease and tobacco etch virus (TEV) protease on Parkin constructs (*Figure 1B*) to analyze the Ubl and RING2 domain rearrangements under native or phosphorylated states. We introduced HRV 3C (between 140th and 141st residue) or TEV (382nd –383rd) sites in the loop regions of Parkin (*Figure 1B*) to avoid any artifacts due to perturbations in native interactions on protein.

First, we tested the ubiquitination activity of Parkin (3C, TEV) to ensure that the inclusion of protease sites did not affect the protein folding or function, which is confirmed by the similar activity of Parkin (3C, TEV) as of the native Parkin construct (*Figure 1—figure supplement 2*). Furthermore, we noticed co-elution of Ubl-linker (1-140) with R0RBR (141-465) in native Parkin (3C, TEV) after treatment with 3C protease, suggesting a stronger interaction between Ubl and the Parkin core (*Figure 1C*). However, in phosphorylated Parkin (3C, TEV) treated with 3C, pUbl-linker (1-140) did not form a complex with R0RBR (141-465), suggesting a poor/no interaction between phospho-Ubl with the core of Parkin (*Figure 1C*). Furthermore, in native Parkin (3C, TEV) treated with TEV, RING2 (383-465) co-eluted with Parkin (1-382), suggesting a stronger interaction between RING2 and the Parkin core in the native Parkin (*Figure 1D*). However, in phospho-Parkin (3C, TEV) treated with TEV, RING2 (383-465) eluted separately from the Parkin (1-382), suggesting that phosphorylation of the Ubl domain results in the displacement of the RING2 domain (*Figure 1D*). All the above data confirmed that the inclusion of molecular scissors on Parkin constructs did not affect Parkin folding. Previous observations that phosphorylation of Ubl weakens Ubl and Parkin interaction, and displacement of RING2 in phospho-Parkin,

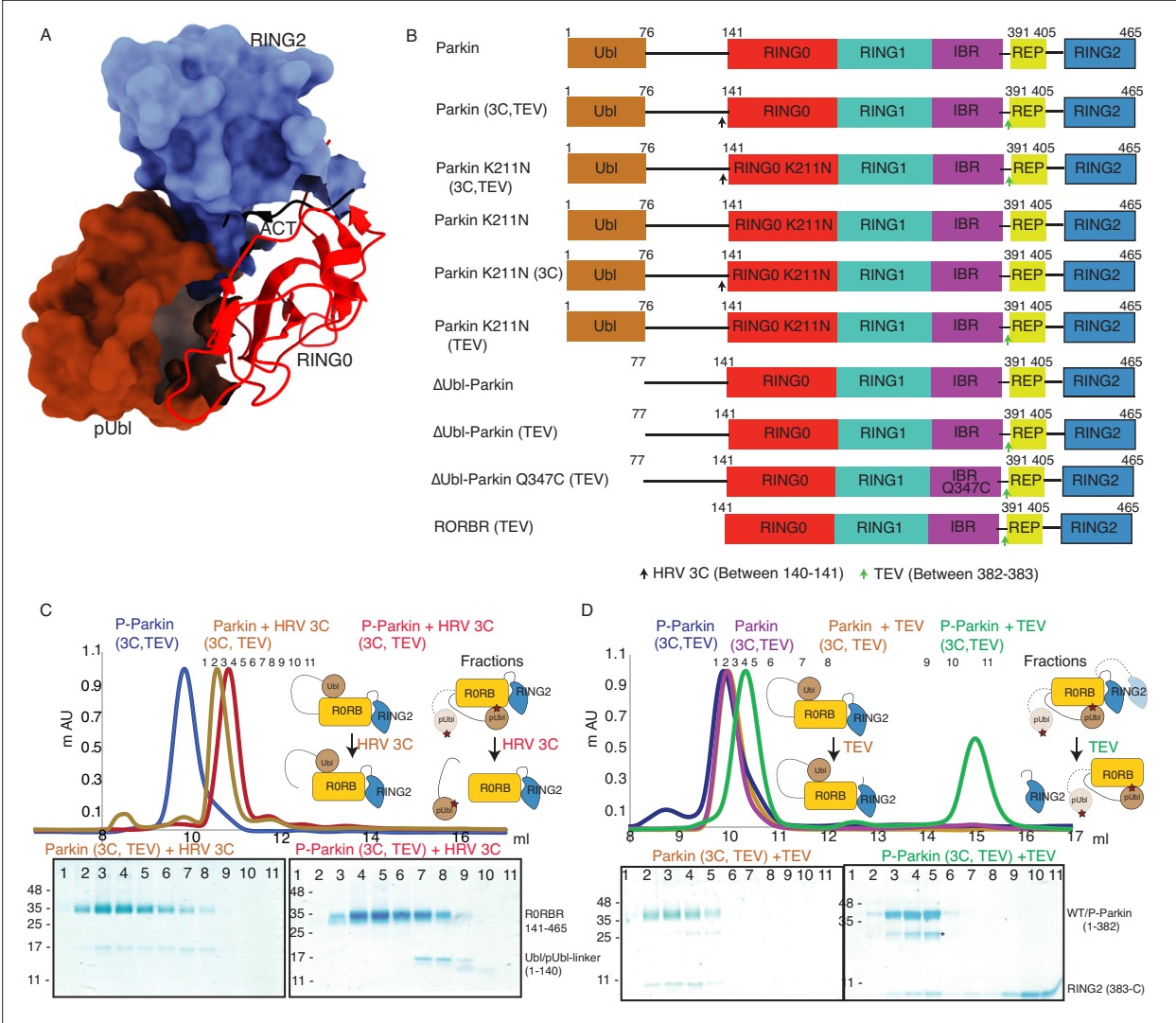

**Figure 1.** Incorporation of molecular scissors to study dynamic conformation upon Parkin phosphorylation. (**A**) Superimposition of WT-Parkin (PDBID: 5C1Z) and phospho-Parkin (PDBID: 6GLC) structures. RING2 (blue), pUbl (brown), RING0 (red), and ACT (black) are shown. For clarity, other Parkin domains are not included. (**B**) Schematic representation of Parkin domains and various constructs used in this study. HRV 3C and TEV sites incorporated in the Parkin construct are marked with black and green arrows, respectively. (**C**) Size-exclusion chromatography (SEC) assay shows the binding/displacement of Ubl-linker (1-140) under native or phosphorylated conditions. A colored key for each trace is provided. Coomassie-stained gels of indicated peaks are shown in the lower panel. A schematic representation is used to explain SEC data. (**D**) Size-exclusion chromatography (SEC) assay shows binding/displacement of RING2 (383-465) under native or phosphorylated conditions. Coomassie-stained gels of indicated peaks are shown in the lower panel. TEV as contamination is indicated (*).

The online version of this article includes the following source data and figure supplement(s) for figure 1:

**Source data 1.** Raw data files used in *Figure 1*.

**Figure supplement 1.** Summary of Parkin conformations/model.

**Figure supplement 2.** Ubiquitination assay to compare Parkin (3C, TEV) or Parkin activity.

**Figure supplement 2—source data 1.** Raw image files.

were validated using our assay. Also, respective proteases only cleaved (untethered) the peptide bond without affecting the native interactions between Parkin domains.

## Phospho-Ubl domain and RING2 domain have a competitive mode of binding on RING0 domain

Previous models of Parkin activation suggested permanent displacement of RING2 after Ubl phosphorylation (*Figure 1—figure supplement 1B*). We wanted to test whether RING2 and pUbl affect the binding of each other on Parkin, which would suggest a competitive binding mode between pUbl and RING2 on the RING0 domain, and a dynamic displaced or bound states of pUbl and RING2. To test the competitive mode of binding between pUbl and RING2 on RING0, and thus affecting the binding of each other, we performed the SEC assay after sequential treatment with HRV 3C and TEV on Parkin (3C, TEV). Interestingly, pUbl-linker (1-140) co-eluted with Parkin core (141-382) upon 3C treatment on fractions that were collected after TEV treatment on phospho-Parkin (3C, TEV) which led to the displacement of RING2 (383-465) (*Figure 2A*). Similarly, RING2 (383-465) co-eluted with Parkin core (141-382) upon TEV treatment on fractions that were collected after 3C treatment on phospho-Parkin (3C, TEV) which led to the displacement of pUbl-linker (1-140) (*Figure 2B*). This data confirmed that pUbl and RING2 competitively bind on RING0. The binding of one negatively affected the binding of the other, unlike previous observations, which only showed phosphorylation of Ubl leading to RING2 displacement.

Our data in *Figure 2B* suggested dynamic displacement of RING2 as untethering of RING2 after pUbl wash-off resulted in stabilization of interactions between RING2 and Parkin core. To further confirm, we crystallized the phospho-Parkin (3C, TEV)-pUb complex after treatment with 3C protease. Treatment with 3C led to displacement of the pUbl-linker (1-140) from the Parkin core (141-465). The overall structure of pUbl-linker (1-140) depleted Parkin (141-465)-pUb complex was determined at 3.3 Å (*Table 1*), and showed similar conformation as seen in previously solved structures of Parkin in the autoinhibited state (*Figure 2C*). The crystal structure showed RING2 bound to RING0, which confirmed that RING2 was only transiently displaced from the RING0 domain in phospho-Parkin and returned to its original position after removal of pUbl-linker (*Figure 2C*), further confirming our SEC data (*Figure 2B*). The crystal structure also revealed that the REP element was bound to the RING1, similar to the autoinhibited state of Parkin (*Figure 2C*). Phospho-ubiquitin was bound to the basic patch between RING0 and RING1 domains, which led to conformational changes in IBR and helix (connecting RING1-IBR domains; *Figure 2C*). In the asymmetric unit, two molecules of Parkin bound to pUb were seen; however, in one of the Parkin molecules, no density was observed in the IBR region (*Figure 2—figure supplement 1*). Overall, this data suggested that pUbl and RING2 exist in a dynamic state in phospho-Parkin (pUbl binding<->RING2 open<->pUbl displaced<->RING2 closed), and compete for binding on RING0.

## K211N mutation on Parkin perturbs RING2 displacement, not pUbl displacement

As phosphorylation of Ubl resulted in the displacement of pUbl from Parkin core (*Figure 1C*), we wondered whether interactions between pUbl and the basic patch (comprising K161, R163, and K211) on RING0 played a key role in pUbl displacement from RING1. Interestingly, similar to phospho-Parkin (3C, TEV) (*Figure 2C*), pUbl-linker (1-140) remained flexible in phospho-Parkin K211N (3C, TEV) and eluted separately from Parkin core (141-465) on SEC (*Figure 3A*). This data suggests that the binding of pUbl with the basic patch on RING0 domain may not be the driving force for pUbl displacement. Further, to confirm that displacement of the RING2 domain is mediated by pUbl binding in the basic patch (K161, R163, and K211) on the RING0 domain, we tested the RING2 displacement using phospho-Parkin K211N (3C, TEV). K211N resulted in stabilization of the RING2 (383-465) domain on phospho-Parkin K211N (1-382) upon TEV treatment, and the two fragments co-eluted on SEC (*Figure 3A*). Although pUbl was displaced in phospho-Parkin K211N, Parkin activity was drastically reduced (*Figure 3B*), suggesting RING2 displacement, not Ubl displacement, is a major cause of Parkin activation. We also noticed a basal level of Parkin activity in the lanes without any activator (pUb), which was reduced in the Parkin K211N mutant (*Figure 3B*). To understand the conformational changes upon mutation in the basic patch on RING0, we also crystallized phospho-Parkin R163D/K211N/Q347C (3C)-pUb complex after treatment with 3C protease, which washed off pUbl-linker

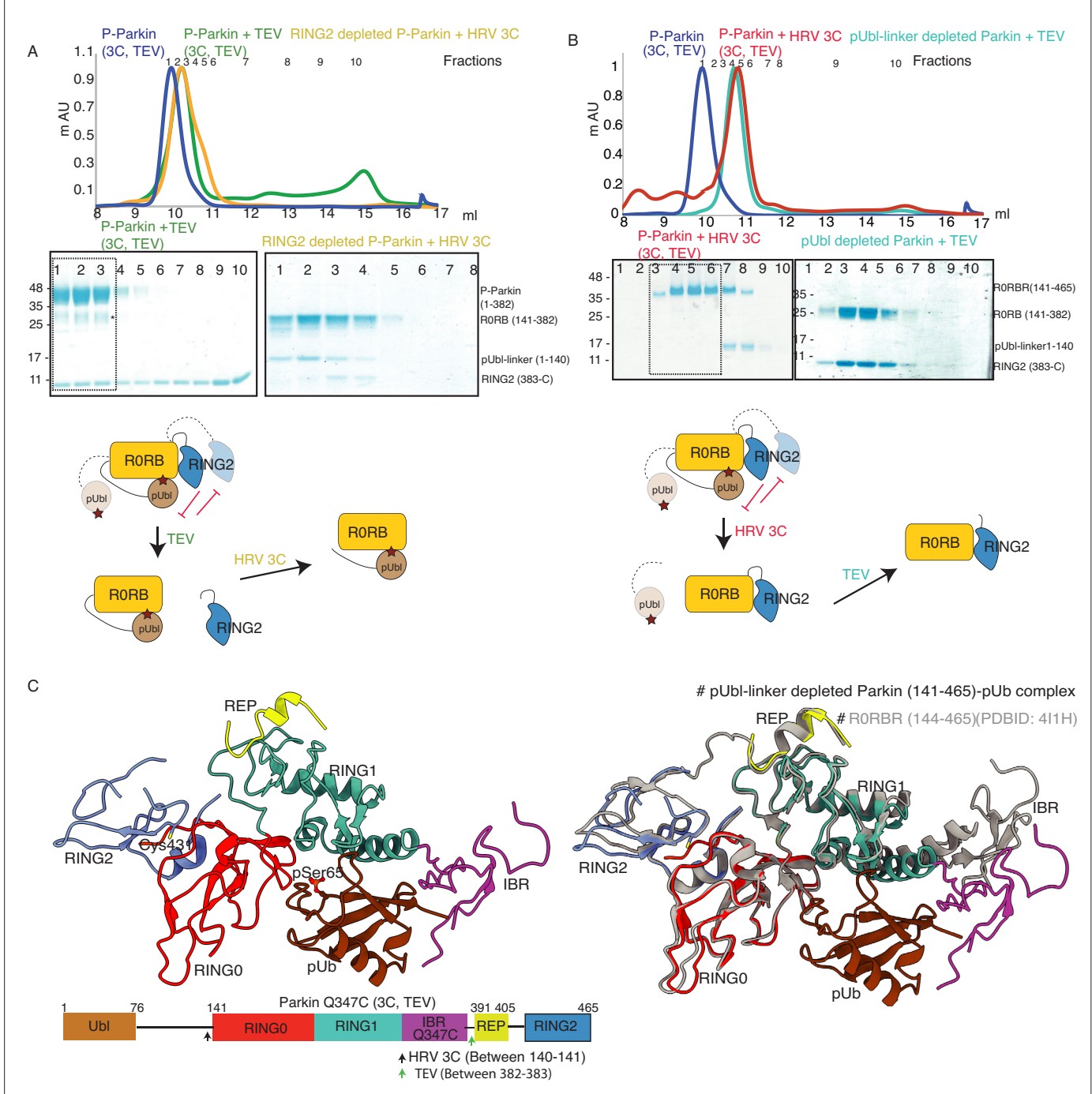

**Figure 2.** Characterization of a competing mode of binding between pUbl and RING2. (**A**) SEC assay shows depletion of RING2 (383-465) from phospho-Parkin stabilize pUbl-linker (1-140) binding with Parkin (141-382) after treatment with 3C protease. Fractions that were pooled for subsequent proteolysis are highlighted in the box. (**B**) SEC assay shows depletion of pUbl-linker (1-140) from phospho-Parkin stabilize RING2 (383-465) binding with Parkin (R0RB, 141–382) after treatment with TEV protease. Fractions that were pooled for subsequent proteolysis are highlighted in the box. (**C**) Crystal structure of pUbl-linker (1-140) depleted Parkin (141-465) complex with pUb (brown). Different domains of Parkin are colored, as shown in the left panel. Catalytic C431 is highlighted. Structure of pUbl-linker (1-140) depleted Parkin (141-465)-pUb complex (colored as in the left panel) is superimposed with R0RBR structure (PDBID: 4I1H, grey) in the right panel. A schematic representation of the Parkin Q347C (3C, TEV) construct used for crystallization is shown at the bottom.

The online version of this article includes the following source data and figure supplement(s) for figure 2:

**Source data 1.** Raw image files.

**Figure supplement 1.** Density map of pUbl-linker (1-140) depleted Parkin (141-465)-pUb complex.

**Table 1.** Data collection and refinement statistics.

| | Ternary trans-complex of phospho-Parkin (1–140 + 141-382 + pUb) | pUbl-linker depleted Parkin (141-465)-pUb complex | Untethered R0RBR | Ternary trans-complex of phospho-parkin with *cis* ACT (1–76 + 77-382 + pUb) | pUbl-linker depleted R0RBR (R163D/K211N)-pUb complex |
|---|---|---|---|---|---|
| **Data collection** | | | | | |
| Resolution range | 34.30–1.92 (1.98–1.92) | 39.15–3.3 (3.41–3.3) | 48.28–2.9 (3.004–2.9) | 35.84–2.6 (2.69–2.6) | 37.47–2.35 (2.43–2.35) |
| Space group | P 32 2 1 | P 64 2 2 | C 2 2 21 | P 32 2 1 | P 1 21 1 |
| Cell dimensions | | | | | |
| *a, b, c* (Å) | 83.804, 83.804, 105.033 | 187.805, 187.805, 141.857 | 86.672, 132.579, 64.692 | 82.764, 82.764, 103.494 | 45.45, 76.426, 114.329 |
| *α, β, γ* (°) | 90, 90, 120 | 90, 90, 120 | 90, 90, 90 | 90, 90, 120 | 90, 100.485, 90 |
| Total reflections | 186235 (17323) | 373278 (38759) | 32244 (3362) | 170918 (17328) | 97886 (9139) |
| Unique reflections | 32991 (3234) | 22705 (2115) | 8492 (843) | 13023 (1273) | 31073 (3105) |
| Multiplicity | 5.6 (5.4) | 16.4 (17.4) | 3.8 (4.0) | 13.1 (13.6) | 3.2 (2.9) |
| Completeness (%) | 99.55 (99.35) | 91.65 (92.19) | 98.84 (99.29) | 99.17 (98.82) | 96.28 (96.61) |
| I/$\sigma$(I) | 14.09 (1.41) | 11.75 (0.72) | 11.83 (3.78) | 17.69 (1.94) | 12.45 (2.55) |
| Wilson B-factor | 43.26 | 132.82 | 49.22 | 46.5 | 40.73 |
| R-merge | 0.05746 (0.9399) | 0.1928 (3.632) | 0.09615 (0.3465) | 0.1416 (1.696) | 0.0735 (0.52) |
| CC1/2 | 0.993 (0.738) | 0.998 (0.48) | 0.993 (0.921) | 0.999 (0.763) | 0.996 (0.695) |
| **Refinement** | | | | | |
| Reflections used in refinement | 32964 (3229) | 20836 (2053) | 8492 (843) | 12933 (1258) | 31062 (3105) |
| R-work/R-free | 0.2031/0.2368 | 0.2360/0.2750 | 0.2180/0.2438 | 0.2141/0.2355 | 0.1952/0.2119 |
| **No of Atoms** | | | | | |
| macromolecules | 2937 | 5457 | 2396 | 3006 | 6033 |
| Ligands | 19 | 59 | 21 | 30 | 37 |
| Solvent | 96 | 2 | 19 | 52 | 258 |
| **RMS deviations** | | | | | |
| Bond length (Å) | 0.008 | 0.009 | 0.008 | 0.009 | 0.007 |
| Bond angles (°) | 1.20 | 1.24 | 1.64 | 1.20 | 1.17 |
| **B-factors** | | | | | |
| macromolecules | 61.01 | 150.78 | 44.37 | 64.69 | 44.87 |
| Ligands | 70.26 | 196.7 | 45.47 | 79.76 | 48.68 |
| Solvent | 60.53 | 118 | 38.72 | 65.63 | 47.36 |
| **Accession code** | 8IKM | 8IK6 | 8JWV | 8IKT | 8IKV |

Data collection and Refinement statistics.

Statistics for the highest-resolution shell are shown in parentheses.

(1-140) from Parkin core (141-465). This complex resulted in better crystals diffracting up to 2.35 Å. The overall structure of the pUbl-linker (1-140) depleted Parkin R163D/K211N/Q347C (141-465)-pUb complex (hereafter R0RBR R163D/K211N-pUb complex) was similar to the autoinhibited structure wherein RING2 was bound on RING0 and REP element was bound on RING1 (*Figure 3C*).

## Untethering of the linker connecting IBR and RING2 allows pUbl binding in *trans*

We next investigated whether the competitive binding between pUbl and RING2 to the RING0 could explain previous reports (*Wenzel et al., 2011*; *Walden and Rittinger, 2018*; *Riley et al., 2013*; *Trempe et al., 2013*; *Wauer and Komander, 2013*; *Wauer et al., 2015*; *Kumar et al., 2015*; *Sauvé et al., 2015*; *Sauvé et al., 2018*; *Condos et al., 2018*; *Gladkova et al., 2018*; *Kumar et al., 2017*) observing the lack of interaction between pUbl and Parkin (lacking Ubl domain) in *trans*. To test this, we used phospho-Parkin K211N, which would not allow the binding of pUbl in the RING0 pocket

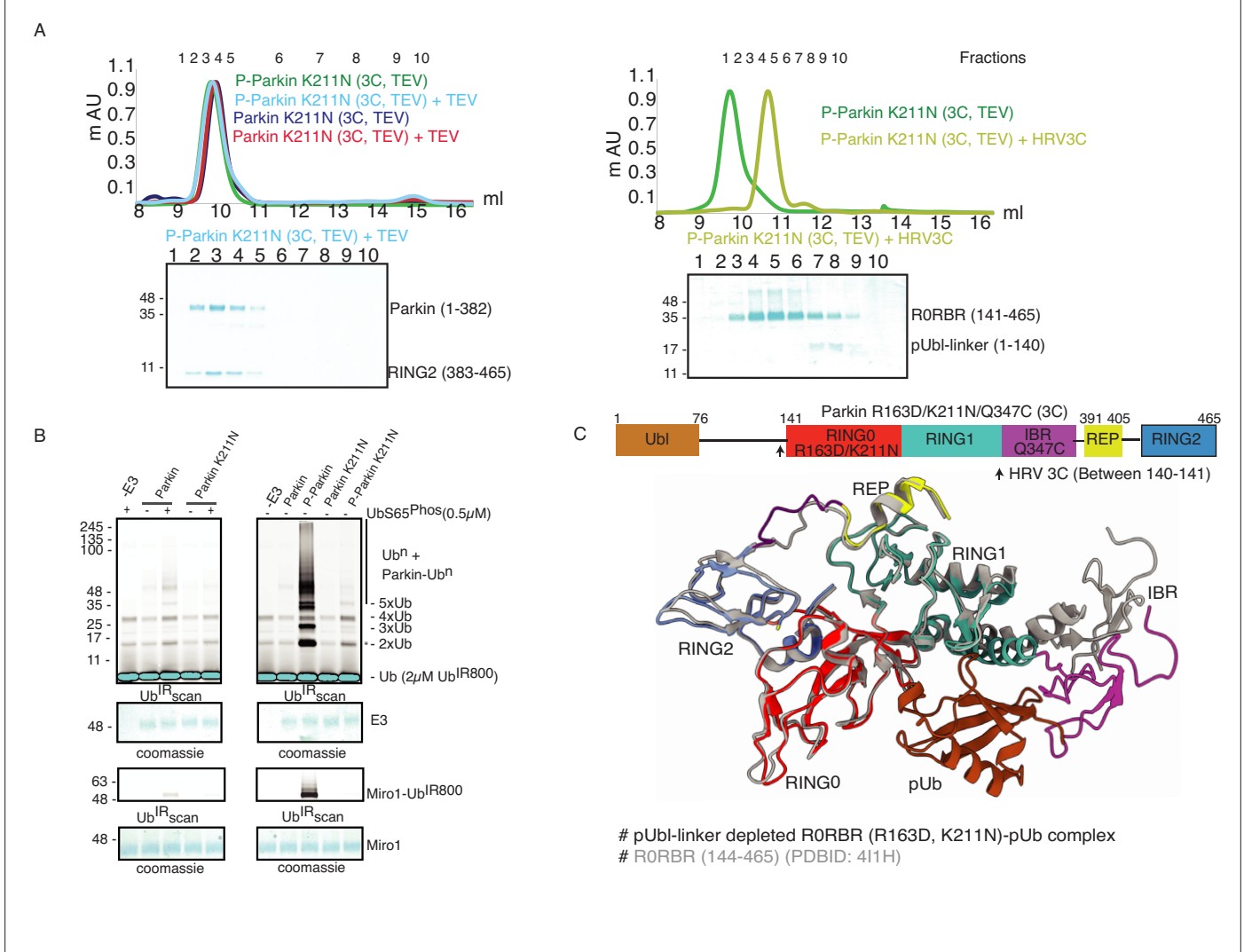

**Figure 3.** K211N mutation affects RING2 displacement, not pUbl. (**A**) Size-exclusion chromatography (SEC) assay to test the displacement of RING2 (left panel) or pUbl-linker (right panel) after phosphorylation of Parkin K211N (3C, TEV). (**B**) Ubiquitination assay to test the activity of Parkin K211N in the presence of pUb or using phospho-Parkin K211N. The middle panel shows a Coomassie-stained loading control. A non-specific, ATP-independent band is indicated (*). The lower panel shows Miro1 ubiquitination for the respective proteins in the upper lane. Coomassie-stained gel showing Miro1 is used as the loading control of substrate ubiquitination assay. (**C**) Crystal structure of pUbl-linker (1-140) depleted R0RBR (R163D/K211N)-pUb complex. The superimposed apo R0RBR structure (PDBID: 4I1H) is shown in grey. A schematic representation of the Parkin R163D/K211N/Q347C (3C) construct used for crystallization is shown at the top.

The online version of this article includes the following source data for figure 3:

**Source data 1.** Raw image files.

of the same molecule, and tested its interaction with ΔUbl-Parkin. However, no complex formation between phospho-Parkin K211N and ΔUbl-Parkin was seen on SEC (*Figure 4A*). We next validated this finding using isothermal titration calorimetry (ITC), which did not show any detectable interaction between phospho-Parkin K211N and ΔUbl-Parkin (*Figure 4A*), consistent with previously published reports.

As our data suggested that the fused domain outcompetes the untethered domain (*Figure 2*), we wondered whether this may explain the lack of detectable binding in *trans*. To test this, we used ΔUbl-Parkin (TEV) treated with TEV as acceptor Parkin, which overcomes the problem of higher net concentration of the fused competing RING2 domain. Acceptor ΔUbl-Parkin (TEV) was treated with TEV, and TEV was removed using an affinity column followed by SEC. SEC showed co-elution of ΔUbl-Parkin (77-382) and RING2 (383-465), confirming that TEV cleaved (untethered) the peptide bond

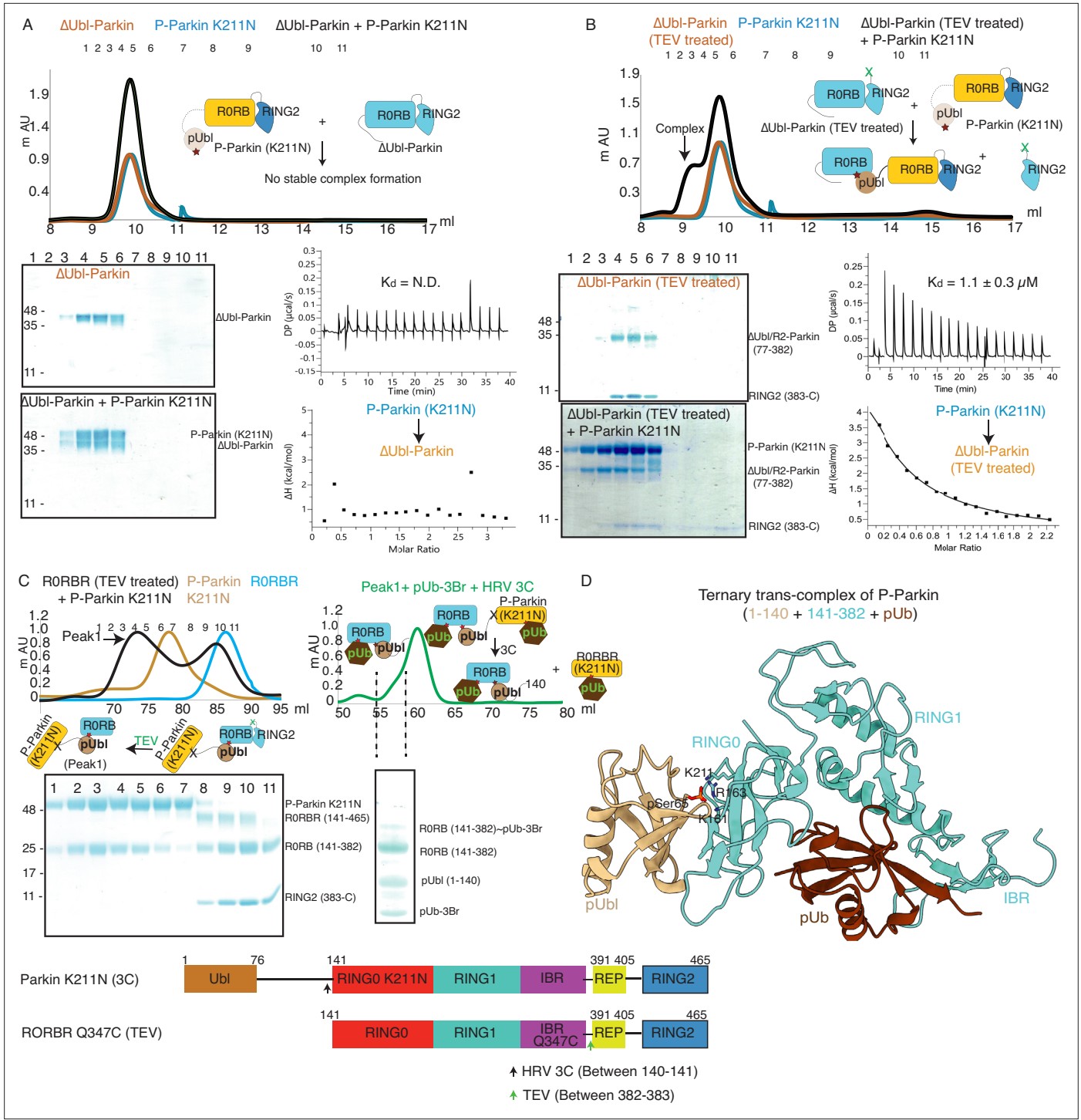

**Figure 4.** Untethering of the linker between IBR-RING2 allows Parkin and phospho-Ubl interaction in *trans*. (**A**) Binding assay between phospho-Parkin K211N and ΔUbl-Parkin. A colored key for each trace is provided. Coomassie-stained gels of indicated peaks are shown in the lower panel. A schematic representation is used to explain SEC data. Isothermal Titration Calorimetry assay between phospho-Parkin K211N and ΔUbl-Parkin is shown in the lower panel. N.D. stands for not determined. (**B**) Binding assay between phospho-Parkin K211N and untethered ΔUbl-Parkin (TEV). A colored key for each trace is provided. Coomassie-stained gels of indicated peaks are shown in the lower panel. A schematic representation is used to explain SEC data. Isothermal Titration Calorimetry assay between phospho-Parkin K211N and untethered ΔUbl-Parkin (TEV) is shown in the lower panel. The dissociation constant (K$_d$) is shown. (**C**) SEC assay to test binding between untethered R0RBR Q347C (TEV) and phospho-Parkin K211N (3C), and displacement of RING2 (383-465) from R0RBR, the left panel. The peak1 (black) containing R0RB (141–382) and phospho-Parkin K211N complex was incubated with pUb-3Br, followed by HRV 3C protease, to purify ternary trans-complex of phospho-Parkin (1–140+141-382 + pUb) on SEC, the right

*Figure 4 continued on next page*

*Figure 4 continued*

panel. The concentrated fractions from the shoulder (highlighted with a dashed line) of the peak in the right panel were loaded on SDS PAGE to confirm complex formation. A schematic representation of the Parkin constructs used for crystallization is shown at the bottom. (D) Crystal structure of the trans-complex of phospho-Parkin with pUb (brown) shows phospho-Ubl domain (wheat) bound to RING0 (cyan) domain of Parkin (cyan).

The online version of this article includes the following source data and figure supplement(s) for figure 4:

**Source data 1.** Raw image files.

**Figure supplement 1.** Parkin treatment with TEV does not affect native interactions.

**Figure supplement 1—source data 1.** Raw image files.

**Figure supplement 2.** Electron density map of the ternary trans-complex of phospho-Parkin.

(connecting IBR and REP-RING2) without affecting the native interactions between ΔUbl-Parkin (77-382) and RING2 (383-465; *Figure 4B*). Incubation of phospho-Parkin K211N with untethered ΔUbl-Parkin (TEV) led to the displacement of RING2 (383-465) from ΔUbl-Parkin (77-382), and a stable trans-complex between phospho-Parkin K211N and ΔUbl-Parkin (77-382) by SEC analysis (*Figure 4B*). The ITC showed a strong affinity ($K_d = 1.1 \pm 0.3$ μM) between phospho-Parkin K211N and untethered ΔUbl-Parkin (TEV; *Figure 4B*), which further supported the SEC data.

Further, to confirm that untethering does not affect the native interactions between RING2 and RING0 domains, we purified and determined the structure of untethered R0RBR (TEV) Parkin (*Figure 4—figure supplement 1A*). Co-elution of R0RB (141-382) and RING2 (383-465) fragments on SEC (*Figure 4—figure supplement 1B*) and crystal structure analysis showing intact native interactions between RING2 and RING0 (*Figure 4—figure supplement 1C*) excluded the possibility of an artifact.

To understand the molecular details of the complex observed in *Figure 4B*, we used Parkin K211N (3C) as a donor of pUbl-linker (1-140) and R0RBR Q347C (TEV) Parkin as an acceptor of pUbl-linker (1-140). Phospho-Parkin K211N (3C) formed a stable complex with untethered R0RBR Q347C (TEV), and RING2 (383-465) was removed from R0RBR Q347C (TEV) (*Figure 4C*). The fractions containing the complex of phospho-Parkin K211N and R0RB (141-382) upon treatment with 3C protease followed by incubation with pUb-3Br showed co-elution of components of the ternary trans-complex (R0RB (141-382), pUbl-linker (1-140), and pUb) on SEC (*Figure 4C*). The crystal structure of the ternary trans-complex of phospho-Parkin (pUbl-linker (1-140)+R0RB (141-382)+pUb) was solved at 1.92 Å (*Table 1*), which further confirmed trans-complex formation between Parkin molecules (*Figure 4D*, *Figure 4—figure supplement 2*). In the crystal structure, the pUbl domain from the donor molecule (phospho-Parkin K211N (3C)) was bound to the basic patch of RING0 on the acceptor molecule (untethered R0RBR (TEV)) (*Figure 4D*) in *trans*. The conformation observed in the trans-complex was similar to the phospho-Parkin (1-382) structure with fused pUbl domain and untethered/truncated RING2 in a *cis* molecule (*Gladkova et al., 2018*; *Sauvé et al., 2018*). Interestingly, the linker connecting pUbl and RING0 remained disordered in all the structures (*Gladkova et al., 2018*; *Sauvé et al., 2018*). Therefore, it would be difficult to say whether, in the previous *cis* structures, the pUbl bound to RING0 was from the same molecule or different molecules. Moreover, the fusion of pUbl with RING0 and untethering/truncation of RING2, as in the earlier structures (*Gladkova et al., 2018*; *Sauvé et al., 2018*), could favor pUbl binding with RING0 in *cis*. Our data established that keeping pUbl and RING2 untethered from their binding partner RING0, thus reducing the artifact due to the higher net concentration of the fused domain with RING0, is ideal for measuring *trans* interactions using biophysical methods.

## Phospho-Parkin activates native Parkin in *trans*

As the pUbl domain remained dynamic in both native phospho-Parkin and phospho-Parkin K211N (*Figure 1C*, *Figure 3A*), we wondered whether a trans-complex was formed between native phospho-Parkin. The latter could also be helpful in the context of activation of various Parkin isoforms lacking either the Ubl domain or RING2 domain (*Figure 5—figure supplement 1*). To test trans-complex formation between native Parkin molecules, we used native phospho-Parkin (1-465) as a pUbl donor and untethered (processed with TEV protease) ΔUbl-Parkin (TEV) as a pUbl acceptor on RING0. Interestingly, phospho-Parkin formed a stable complex with ΔUbl-Parkin (77-382) and RING2 (383-465) was

displaced from untethered ΔUbl-Parkin (TEV) (*Figure 5A*, *Figure 5—figure supplement 2A*), similar to the interaction between phospho-Parkin K211N and untethered ΔUbl-Parkin (TEV).

We further tested the binding of phospho-Parkin with untethered WT-Parkin (TEV). Similar to untethered ΔUbl-Parkin (TEV), untethered WT-Parkin (TEV) formed a complex with phospho-Parkin, and resulted in the removal of RING2 (383-465) from WT-Parkin (1-382) (*Figure 5C*, *Figure 5—figure supplement 2B*). However, unlike untethered WT-Parkin (TEV), untethered Parkin K211N (TEV) failed to form the complex with phospho-Parkin (*Figure 5C*, *Figure 5—figure supplement 2C*). This latter finding confirmed that interactions between pUbl and the basic patch on the RING0 domain form a trans-complex. To further validate this, we also confirmed complex formation using SEC-MALS (size-exclusion chromatography coupled with multi-angle light scattering). MALS analysis further confirmed complex (Phospho-Parkin and WT-Parkin (1-382), Observed M. W.=94 ± 3 Kda) formation between phospho-Parkin (Observed M. W.=53 ± 2 Kda) and untethered WT-Parkin (TEV) (Observed M. W.=52 ± 3 Kda) (*Figure 5D*, *Figure 5—figure supplement 2B*).

As our binding experiments suggested interaction between phosphorylated Parkin and native Parkin, we next checked whether phosphorylated Parkin can activate native Parkin. To test phospho-Parkin mediated Parkin activation in *trans*, we used a catalytic-inactive version of phospho-Parkin T270R/C431A with mutations in both the E2 binding site (T270R) and catalytic site (C431A). Interestingly, we observed that WT-Parkin ubiquitination/autoubiquitination activity was increased with increasing concentrations of phospho-Parkin T270R/C431A (*Figure 5E*). Although, we were not expecting activation of WT-Parkin by phospho-Parkin as Ubl of WT-Parkin would block the E2 binding site on RING1 in WT-Parkin, activation of WT-Parkin with phospho-Parkin T270R/C431A suggested that a significant inhibition on Parkin is mediated by RING0 blocking RING2, which was released upon pUbl binding.

Further, we wondered whether pUbl would enhance Parkin phosphorylation similar to pUb (*Kazlauskaite et al., 2015*). To test this, we checked Parkin phosphorylation by PINK1 in the presence of pUbl or pUb. However, unlike pUb, pUbl did not enhance Parkin phosphorylation by PINK1 (*Figure 5—figure supplement 2D*), confirming that pUbl and pUb binding lead to unique conformational changes in Parkin. Overall, this data demonstrates pUbl-mediated dimerization of Parkin molecules leading to Parkin activation in *trans*.

## Assessment of Parkin activation in cells

It has previously been reported that pUb may interact with the RING0 domain of Parkin and that loss of this interaction underlies loss of Parkin recruitment to the mitochondria in cells expressing Parkin K211N (*Tang et al., 2017*). However, we recently showed that pUb does not bind in the RING0 pocket comprising K161, R163, and K211, and pUb binds specifically in the RING1 pocket comprising K151, R305, and H302 (*Lenka et al., 2023*), unlike phospho-Ubl binding in the RING0 pocket and displacing RING2 in *trans* (*Figure 5*). Biophysical assays also revealed that unlike the tight binding of pUb in the RING1, pUbl binding in the RING0 pocket was very transient. Furthermore, K211N mutation in the RING0 pocket resulted in a loss of Parkin activity by both loss of pUbl-mediated interactions (*Figure 5*) and by N211-driven conformational changes leading to loss of Parkin activity independent of pUb binding (*Lenka et al., 2023*). This loss of Parkin activity would lead to a reduced amount of pUb, resulting in loss of Parkin recruitment to mitochondria. Therefore, we decided to test an activity-independent Parkin recruitment to impaired mitochondria using a Parkin translocation assay in HeLa cells (*Kane et al., 2014*; *Shiba-Fukushima et al., 2014*; *Shiba-Fukushima et al., 2012*; *Ordureau et al., 2014*; *Lazarou et al., 2013*). Consistent with previous studies, (*Kane et al., 2014*; *Shiba-Fukushima et al., 2014*; *Shiba-Fukushima et al., 2012*; *Ordureau et al., 2014*; *Lazarou et al., 2013*) full-length wild-type but not catalytic-inactive GFP-Parkin C431F was recruited to mitochondria following carbonyl cyanide m-chlorophenyl hydrazone (CCCP) treatment (*Figure 6—figure supplement 1A*, B). Similarly, we did not observe the recruitment of GFP-Parkin C431F/H302A or GFP-Parkin C431F/K211N mutants to impaired mitochondria when expressed alone (*Figure 6—figure supplement 1A*, B).

We observed that co-expression of mCherry-tagged-Parkin WT with GFP-Parkin C431F enabled GFP-Parkin C431F recruitment to the mitochondria, similar to a previous study (*Lazarou et al., 2013*; *Figure 6A and D*). Under these assay conditions, we strikingly observed that mutation of the pUb binding pocket in the RING1 completely abolished recruitment of the double mutant GFP-Parkin

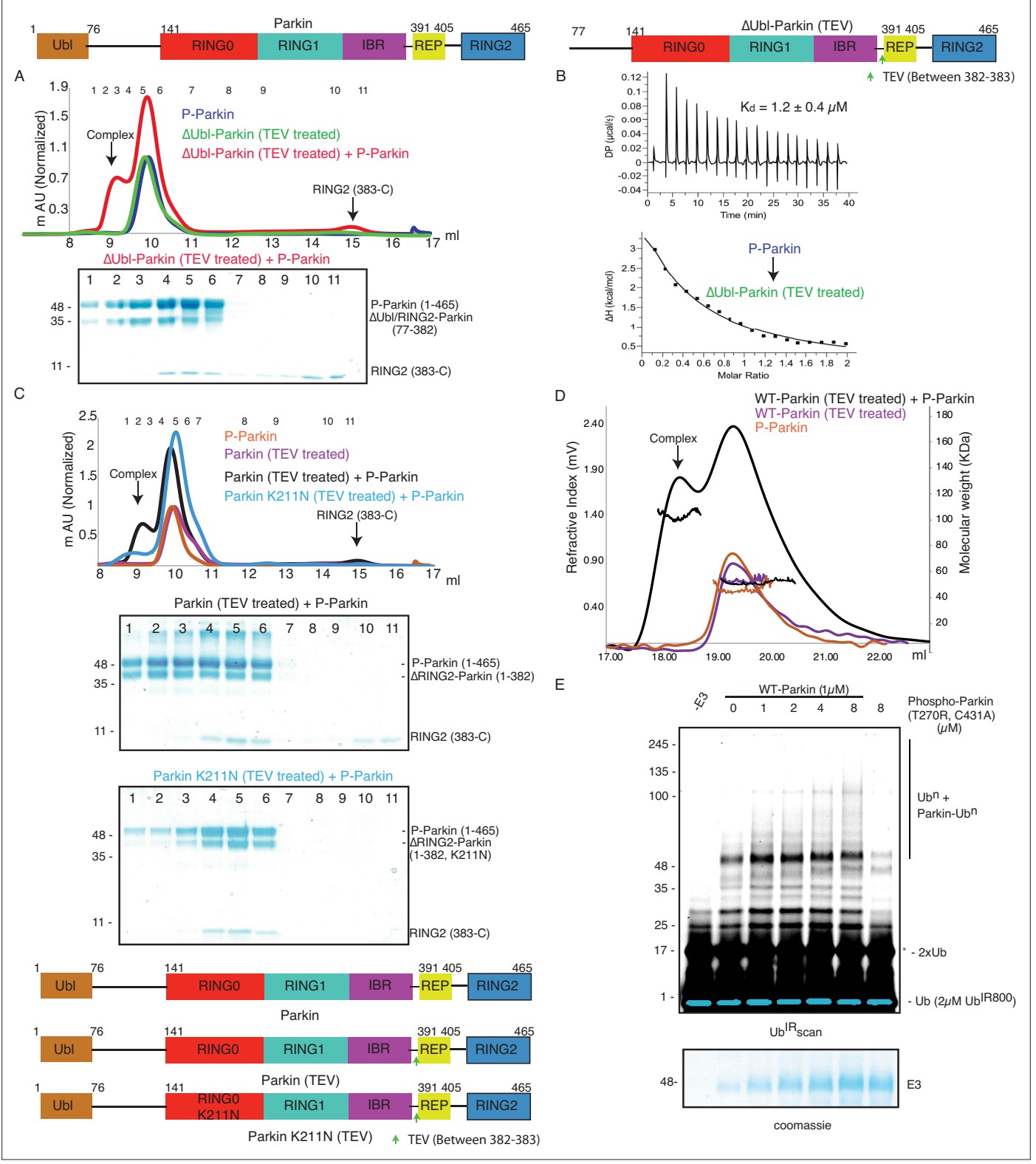

**Figure 5.** Parkin dimerization and trans-activation of native Parkin are mediated by phosphorylation of the Ubl domain of Parkin. (**A**) SEC assay between phospho-Parkin and untethered ΔUbl-Parkin. A colored key for each trace is provided. Coomassie-stained gels of indicated peaks are shown in the lower panel. TEV protein contamination is indicated (*). A schematic representation of the Parkin constructs used for experiments in panels A and B is shown at the top. (**B**) Isothermal Titration Calorimetry assay between phospho-Parkin and untethered ΔUbl-Parkin (TEV). The dissociation constant ($K_d$) is shown (**C**) SEC assay between phospho-Parkin and untethered WT-Parkin (TEV) (upper panel) or untethered Parkin K211N (TEV) (lower panel).

*Figure 5 continued on next page*

*Figure 5 continued*

A schematic representation of the Parkin constructs used for experiments in panels C and D is shown at the bottom. (**D**) SEC-MALS assay to confirm the complex formation between untethered WT-Parkin (TEV) and phospho-Parkin. (**E**) Ubiquitination assays to check the WT-Parkin activation (right panel) with increasing concentrations of phospho-Parkin T270R/C431A. A non-specific, ATP-independent band is indicated (*). The lower panel shows a Coomassie-stained loading control.

The online version of this article includes the following source data and figure supplement(s) for figure 5:

**Source data 1.** Raw image files.

**Figure supplement 1.** Schematic representation of domain organization in various isoforms (*La Cognata et al., 2018*; *Scuderi et al., 2014*) of Parkin.

**Figure supplement 2.** Phosphorylation of Parkin leads to the association of Parkin molecules in *trans*.

**Figure supplement 2—source data 1.** Raw image files.

C431F/H302 to the mitochondria when co-expressed with mCherry-tagged-Parkin WT (*Figure 6B and D*). This excluded a significant role for the RING0 pocket in pUb binding in the context of full-length parkin expressed in cells following mitochondrial damage (*Figure 6B and D*). In line with this, mutation of the RING0 binding pocket produced a moderate defect in recruitment of the double mutant GFP-Parkin C431F/K211N to the mitochondria when co-expressed with mCherry-tagged-Parkin WT (*Figure 6C and D*), suggesting that the transient interaction between pUbl and RING0 of Parkin in *trans* acts in concert with pUb binding to RING1 pocket for optimal Parkin recruitment to sites of mitochondrial damage (*Figure 6C and D*). Under all transfection conditions, we did not observe a significant difference in mCherry-tagged Parkin WT (*Figure 6—figure supplement 1C*). Furthermore, co-expression of GFP-Parkin C431F or GFP-Parkin C431F/K211N or GFP-Parkin C431F/H302A with the non-phosphorylatable mCherry-tagged-Parkin S65A failed to rescue recruitment to the mitochondria (*Figure 6A–D*). These findings were in line with our biophysical data and highlight the importance of phospho-Ubl domain-mediated interactions in Parkin recruitment to the mitochondria.

## ACT improves enzyme kinetics of Parkin

A previous study identified a small region (101-109) in the linker between Ubl and RING0 as an activator element (ACT) required for Parkin activity (*Gladkova et al., 2018*). To further explore the role of the ACT, we tested whether the omission of ACT affects the binding of Parkin with the charged state of E2 (E2~Ub). We observed a tight complex formation between phospho-Parkin, pUb, and E2~Ub on SEC assay (*Figure 7A*). Interestingly, deletion of the ACT did not affect the complex formation with E2~Ub, as phospho-Parkin ΔACT co-eluted with pUb and E2~Ub (*Figure 7A*). As the displacement of RING2 is a crucial process during Parkin activation, we tested whether the removal of the ACT affects the displacement of the RING2 domain using our TEV-based SEC assay. We observed that phospho-Parkin ΔACT (TEV) after treatment with TEV resulted in a shift where RING2 (383-465) was displaced from the Parkin core (1–382, ΔACT), resulting in the elution of two fragments of Parkin separately on SEC (*Figure 7B*). As the deletion of ACT did not show any functional defect in Parkin, we hypothesized that the presence of ACT at the interface of RING0 and RING2 might affect the dynamic nature of RING2, thereby regulating the enzyme kinetics. To test this hypothesis, we compared the phospho-Parkin ΔACT ubiquitination activity over different time points. We observed that the deletion of ACT slowed the kinetics of Parkin activity, doubling the time for phospho-Parkin ΔACT to reach a similar level of activity as phospho-Parkin (*Figure 7C*).

## ACT is more efficient in *cis*

The ternary trans-complex of phospho-Parkin (1–140+141-382 + pUb) structure in this study was solved at a similar resolution and in the same space group as the previously solved structure of phospho-Parkin (1-382) in complex with pUb (*Gladkova et al., 2018*). In the previous structure of phospho-Parkin (1-382)-pUb complex (PDBID: 6GLC), the ACT region was clearly shown to occupy the hydrophobic pocket on RING0 (*Figure 8A*). However, we did not see any density of the ACT region in the ternary trans-complex structure of phospho-Parkin (1–140+141-382 + pUb) (*Figure 8A*, *Figure 8—figure supplement 1A*). Interestingly, we observed that in the ternary trans-complex structure of phospho-Parkin, K48 of the pUbl domain occupied the same pocket that R104 of the ACT region occupied in the structure of phospho-Parkin-pUb complex (*Figure 8A*, *Figure 8—figure*

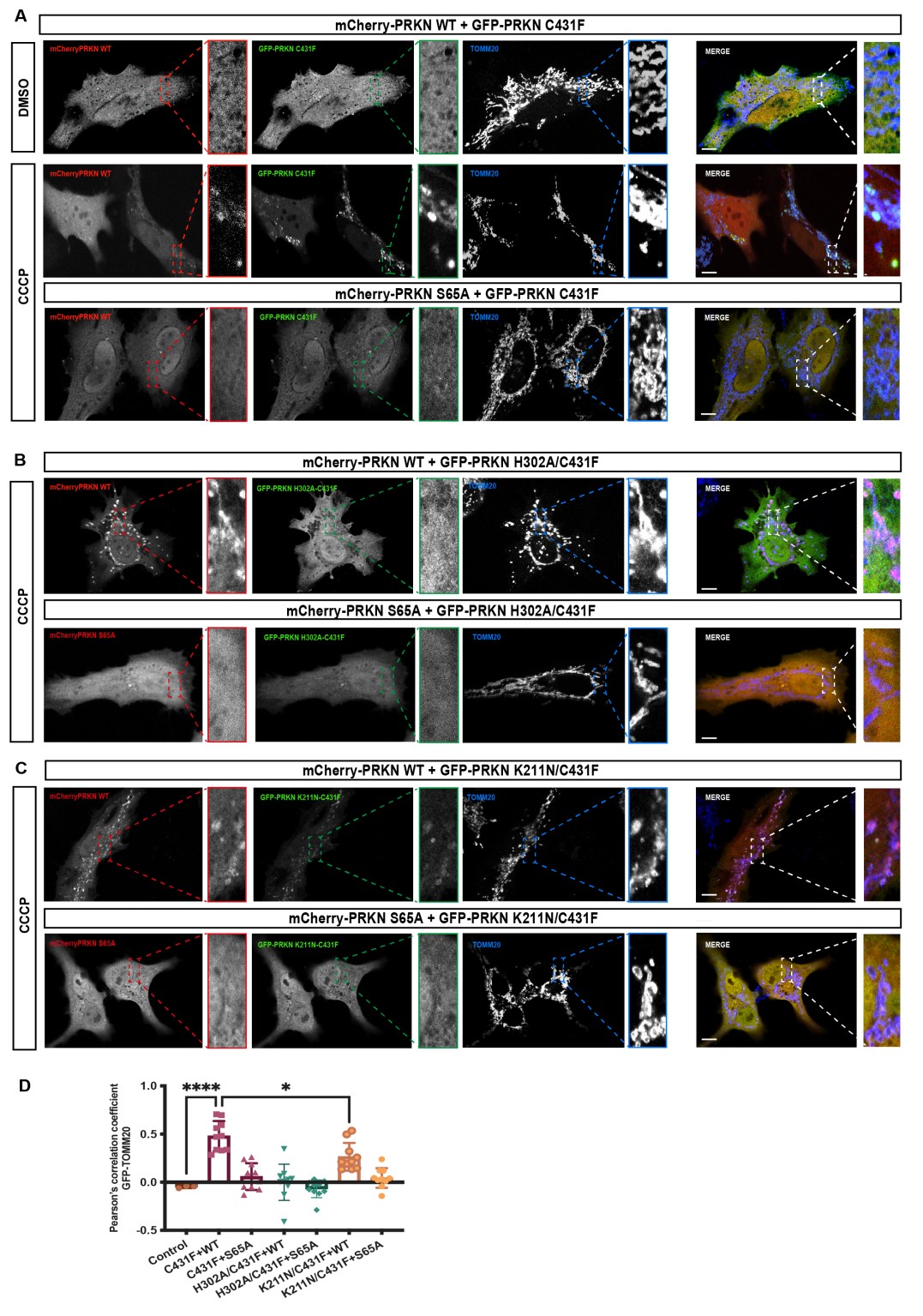

**Figure 6.** Analysis of Parkin mutant recruitment to mitochondria in HeLa cells. (**A**) Immunofluorescence of HeLa cells co-transfected with either mCherry-Parkin wild-type (WT) or mCherry-Parkin S65A and GFP-Parkin C431F or, (**B**) GFP-Parkin H302A/C431F and, (**C**) GFP-Parkin K211N/C431F. Cells were treated for 1 hr with 10 μM CCCP, and DMSO was used as a control. Mitochondria were labeled with anti-TOMM20 antibody (blue). Scale bar = 10 μm. (**D**) Quantification of GFP- Parkin (WT and mutants) on mitochondria. The co-localization of GFP-Parkin (WT and mutants) with TOMM20 (mitochondria) was evaluated using Pearson's correlation coefficient. Errors are represented as S.D. Statistical differences in Pearson's correlation coefficient were evaluated using one-way ANOVA and Tukey's multiple comparisons post-test. Statistical significance is as follows: *, $p < 0.05$; ****, $p < 0.0001$.

*Figure 6 continued on next page*

*Figure 6 continued*

The online version of this article includes the following figure supplement(s) for figure 6:

**Figure supplement 1.** Parkin localization on mitochondria.

*supplement 1A*). Also, the side-chain of K48 of the pUbl domain was disordered in the previous structure of the phospho-Parkin (1-382)-pUb complex (*Figure 8A*).

We wondered whether the lack of density in the ACT region was due to the preference of ACT to remain associated with the *cis* molecule rather than to be complemented by the *trans* molecule. To test this hypothesis, we determined the crystal structure of the ternary trans-complex of phospho-Parkin with *cis* ACT using phospho-Ubl (1-76) and ΔUbl-Parkin Q347C (TEV). pUbl formed a stable complex with untethered ΔUbl-Parkin Q347C (TEV) and resulted in the displacement of RING2 (383-465) (*Figure 8B*). Fractions containing trans-complex of phospho-Parkin (1–76+77-382) with *cis* ACT were mixed with pUb-3Br to get the crystals of the ternary complex. The ternary trans-complex of phospho-Parkin (1–76+77-382 + pUb) with *cis* ACT was crystallized, and structure was determined at 2.6 Å (*Table 1*). Interestingly, in the structure of the ternary trans-complex of phospho-Parkin with *cis* ACT, we could observe the electron density of the ACT region (*Figure 8C*, *Figure 8—figure supplement 1B*). Furthermore, K48, which occupied the ACT region in the ternary trans-complex structure

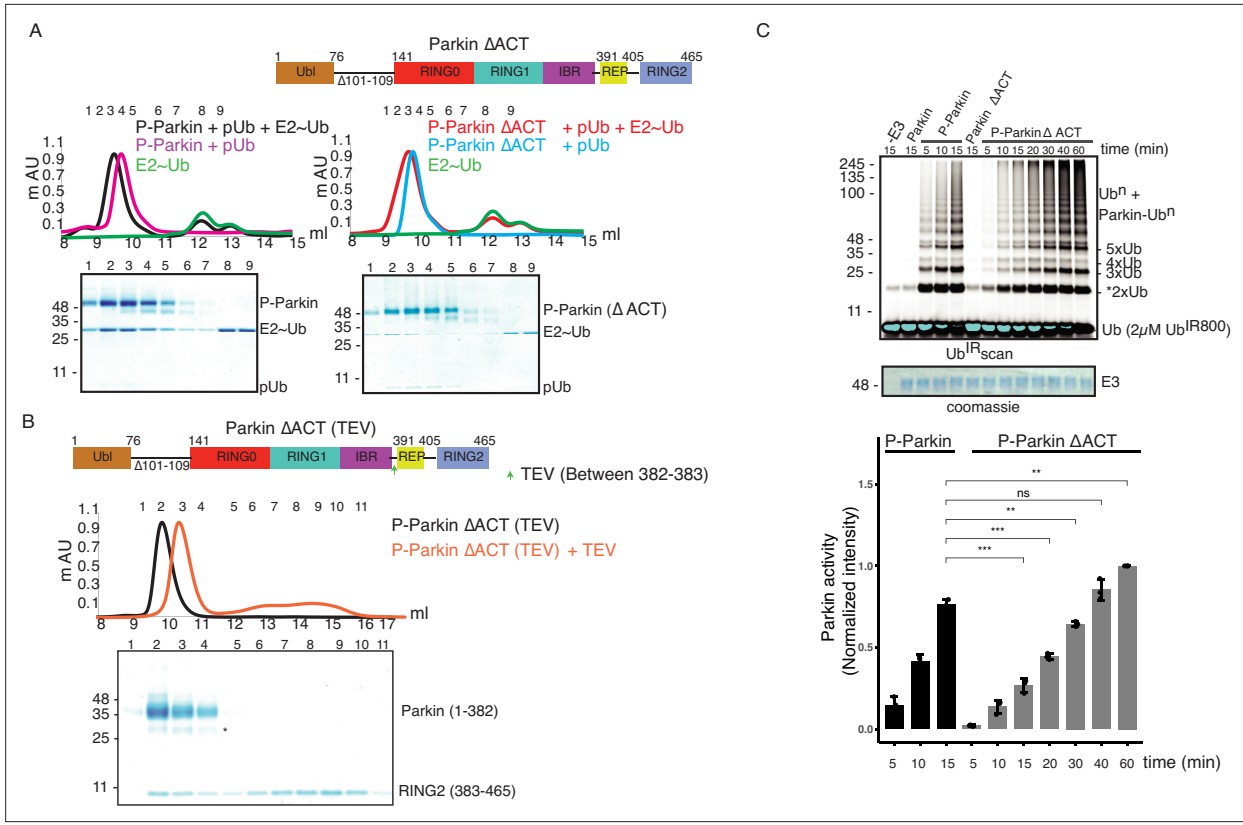

**Figure 7.** ACT plays a crucial role in enzyme kinetics. (**A**) Size-exclusion chromatography (SEC) assay to test the binding of E2~Ub$_{don}$ with phospho-Parkin (left panel) or phospho-Parkin ΔACT (right panel). Assays were done using Parkin in a complex with pUb. A colored key for each trace is provided. Coomassie-stained gels of indicated peaks are shown in the lower panel. The upper panel shows a schematic representation of the Parkin ΔACT construct used. (**B**) Size-exclusion chromatography (SEC) assay to check displacement of the RING2 domain after phosphorylation of Parkin ΔACT. The upper panel shows a schematic representation of the Parkin ΔACT construct used for the RING2 displacement assay. Conformational changes in Parkin, as observed by the SEC experiment, are shown schematically. (**C**) Ubiquitination assay to check the effect of ACT deletion (ΔACT) on Parkin activity. A non-specific, ATP-independent band is indicated (*). The middle panel shows a Coomassie-stained loading control. In the lower panel, the bar graph shows the integrated intensities of ubiquitin levels from three independent experiments (mean ± s.e.m.). Statistical significance was determined using pair-wise student's t-test (**p<0.01, ***p<0.001, ns-nonsignificant).

The online version of this article includes the following source data for figure 7:

**Source data 1.** Raw image files.

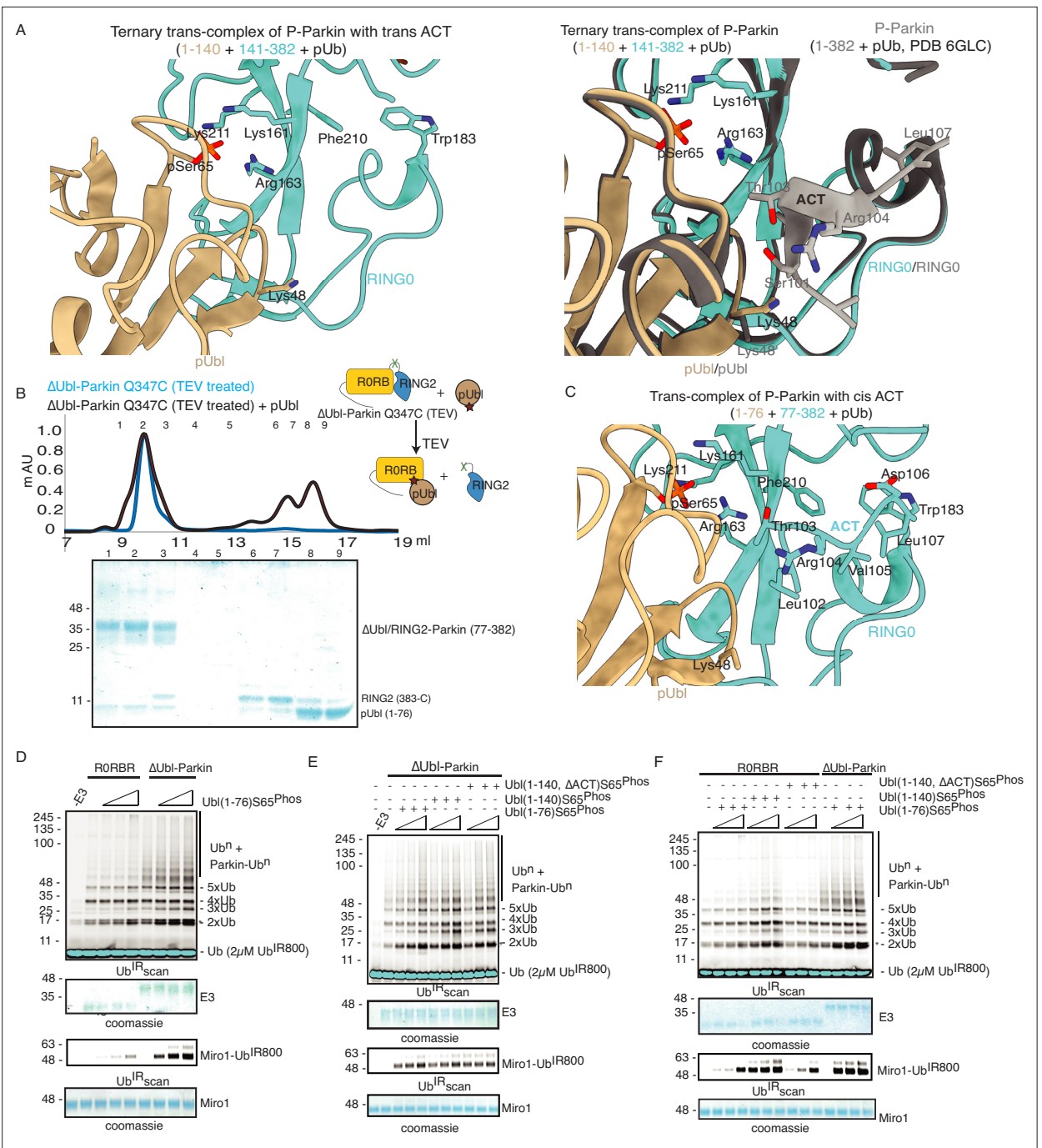

**Figure 8.** ACT is more efficient in *cis*. (**A**) Crystal structure of ternary trans-complex of phospho-Parkin with pUb (1–140+141-382 + pUb), left panel. pUbl (wheat) and RING0 (cyan) of Parkin are shown. The right panel shows superimposed structures of ternary trans-complex of phospho-Parkin with pUb, colored as the left panel, and the phospho-Parkin complex with pUb (PDBID: 6GLC) is shown in grey. (**B**) SEC assay to check the binding between untethered ΔUbl-Parkin (TEV) and phospho-Ubl (1-76). A colored key for each trace is provided. Coomassie-stained gels of indicated peaks are shown in the lower panel. (**C**) Crystal structure of ternary trans-complex of phospho-Parkin with *cis* ACT (1–76+77-382 + pUb) shows ACT (cyan) present in the pocket on RING0 (Cyan) and pUbl (wheat) in the vicinity. (**D**) Comparison of R0RBR and ΔUbl-Parkin activation using the increasing concentrations of pUbl (1-76). A non-specific, ATP-independent band is indicated (*). The middle panel shows a Coomassie-stained loading control. The lower panel shows Miro1 ubiquitination for the respective proteins in the upper lane. Coomassie-stained gel showing Miro1 is used as the loading control of substrate ubiquitination assay. (**E**) Ubiquitination assay of ΔUbl-Parkin with increasing concentrations of pUbl (1-76), pUbl-linker (1-140), pUbl-linker-ΔACT (1–140, Δ101–109). A non-specific, ATP-independent band is indicated (*). The middle panel shows a Coomassie-stained loading control. The lower panel shows Miro1 ubiquitination for the respective proteins in the upper lane. Coomassie-stained gel showing Miro1 is used as the loading control of substrate ubiquitination assay. (**F**) Comparison of R0RBR and ΔUbl-Parkin activation using the increasing concentrations of pUbl (1-76)/pUbl-linker (1-

*Figure 8 continued on next page*

*Figure 8 continued*

140)/pUbl-linker-ΔACT (1–140, Δ101–109). A non-specific, ATP-independent band is indicated (*). The middle panel shows a Coomassie-stained loading control. The lower panel shows Miro1 ubiquitination for the respective proteins in the upper lane. Coomassie-stained gel showing Miro1 is used as the loading control of substrate ubiquitination assay.

The online version of this article includes the following source data and figure supplement(s) for figure 8:

**Source data 1.** Raw image files.

**Figure supplement 1.** Role of ACT in Parkin activation.

**Figure supplement 1—source data 1.** Raw image files.

of phospho-Parkin with *trans* ACT, was disordered in the ternary trans-complex structure of phospho-Parkin with *cis* ACT, similar to what was seen previously in the phospho-Parkin structure (*Figure 8A and C*, *Figure 8—figure supplement 1B*).

To validate crystal structures, we compared the ubiquitination activity of R0RBR (141-465) and ΔUbl-Parkin (77-465) in the presence or absence of pUb. The presence of a linker (77-140) containing ACT in ΔUbl-Parkin (77-465) made it more active compared to R0RBR (141-465) (*Figure 8—figure supplement 1C*). We then compared the activation of R0RBR and ΔUbl-Parkin using pUbl (1-76) in *trans*. We observed that pUbl (1-76) efficiently activated ΔUbl-Parkin (77-465); however, R0RBR (141-465) activation by pUbl (1-76) was very poor (*Figure 8D*, *Figure 8—figure supplement 1D*). Further, we tested whether pUbl-linker (1-140) with or without ACT would affect the activation of ΔUbl-Parkin (77-465) in *trans*. Interestingly, ubiquitination assays performed using increasing concentrations of pUbl (1-76) or pUbl-linker (1-140), or pUbl-linker-ΔACT (1–140, Δ101–109) showed that ΔUbl-Parkin activation was not affected by the linker (77-140) or ACT region in *trans* (*Figure 8E*). However, compared to pUbl (1-76), pUbl-linker (1-140) showed better activation of R0RBR (141-465) (*Figure 8F*, *Figure 8—figure supplement 1E*). Also, in contrast to pUbl-linker (1-140), pUbl-linker-ΔACT (1–140, Δ101–109) showed poor activation of R0RBR (141-465) which was similar to pUbl (1-76) (*Figure 8F*, *Figure 8—figure supplement 1E*). However, the activity of R0RBR (141-465) complemented with pUbl-linker (1-140) was less than the activity of ΔUbl-Parkin (77-465) complemented with pUbl (1-76) (*Figure 8F*, *Figure 8—figure supplement 1E*). Overall, our data suggested that ACT can be complemented in *trans*; however, ACT is more efficient in *cis*.

## Crystal structure of pUbl-linker (1-140) depleted R0RBR (R163D/K211N)-pUb complex reveals a new ubiquitin-binding site on Parkin

In the last few years, several structures of Parkin or Parkin complexes were solved in various conditions and from different species. However, the linker (408-415) between REP element and RING2 was mostly disordered, except in structures (PDBID: 4I1H, 5CAW, 4ZYN) where the above region was modeled in different conformations (*Figure 9—figure supplement 1A*), highlighting its flexible nature. A pathogenic mutation T415N was also found in the linker (408-415), which abolished Parkin activity. However, the role of this small linker region on Parkin remains elusive. Therefore, we decided to inspect all the structures solved in the present study. In the crystal structure of R0RBR (R163D/K211N)-pUb complex, out of two molecules of Parkin in the asymmetric unit, one molecule of Parkin showed nice electron density of the linker (408-415) region of Parkin (*Figure 9A and B*). We further noticed conformational changes in the linker (408-415) region in the structure of the R0RBR (R163D/K211N)-pUb complex when compared to the previously solved apo R0RBR structure (PDBID: 4I1H) (*Figure 9—figure supplement 1B*). While T410, I411, and K412 were facing outwards in the apo R0RBR structure, in the structure of R0RBR (R163D/K211N)-pUb complex these residues were present in the core (*Figure 9B*, *Figure 9—figure supplement 1B*). Interestingly, we noticed interactions between the linker (408-415) of Parkin and pUb from the neighboring molecule of the asymmetric unit (*Figure 9C*). The core of interactions between the Parkin linker and ubiquitin was mediated by I411, which was involved in hydrophobic interactions with the hydrophobic pocket of ubiquitin (*Figure 9C*). Other interactions between Parkin and ubiquitin included ionic interactions mediated by K412, and H422 (*Figure 9C*). Water-mediated interactions between linker (408-415) and ubiquitin included T410 with the carbonyl group of R72 of ubiquitin, and T415 with the carbonyl of G35 of ubiquitin (*Figure 9C*). Furthermore, E409 formed a salt-bridge with K413 (*Figure 9C*), which could be required for maintaining the structure of the linker region for ubiquitin binding. Also, residues in

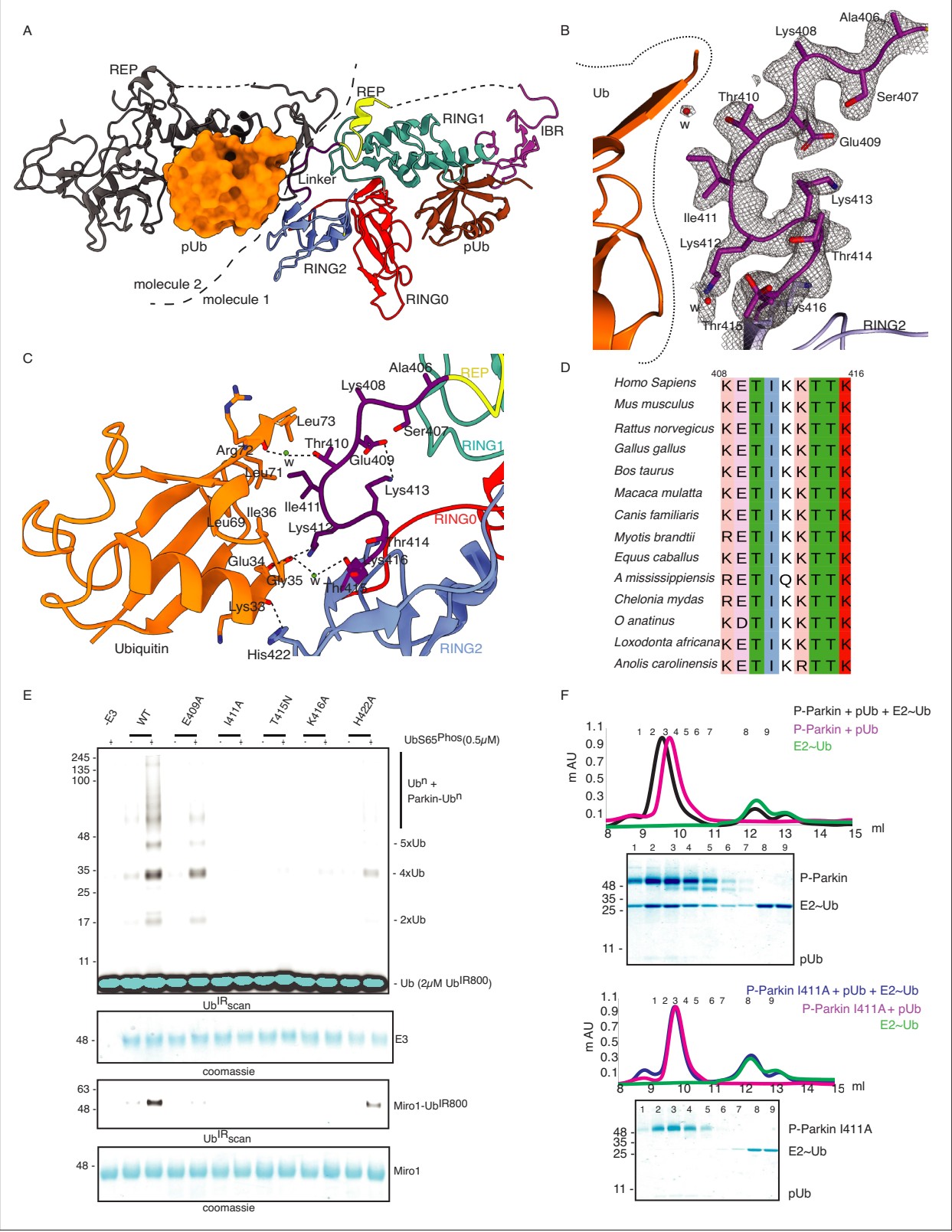

**Figure 9.** Linker (408-415) of Parkin binds with donor ubiquitin ($Ub_{don}$) of E2-$Ub_{don}$. (**A**) The asymmetric unit of the crystal structure of pUbl-linker (1-140) depleted R0RBR (R163D/K211N)-pUb complex. Parkin molecule-1 (domains are shown in different colors) and pUb (brown) are shown. Parkin molecule-2 (grey) and pUb (orange) are shown. The interface of two Parkin molecules is highlighted (dashed line). (**B**) The 2Fo-Fc map (grey) of the linker region between REP and RING2. 2Fo-Fc map is contoured at 1.5 σ. Water molecules are represented as w. (**C**) Crystal structure shows interactions between

*Figure 9 continued on next page*

*Figure 9 continued*

the linker (408-415) and ubiquitin. Different regions are colored as in panel A. Hydrogen bonds are indicated as dashed lines. (**D**) Sequence alignment of Parkin from various species highlighting conservation in the linker (408-415) region. Residue numbers shown on top of sequence alignment are according to human Parkin. (**E**) Ubiquitination assay of Parkin mutants in the linker region. The middle panel shows a Coomassie-stained loading control. The lower panel shows Miro1 ubiquitination for the respective proteins in the upper lane. Coomassie-stained gel showing Miro1 is used as the loading control of substrate ubiquitination assay. (**F**) Size-exclusion chromatography (SEC) assay to compare the binding of E2~Ub with phospho-Parkin (upper panel) or phospho-Parkin I411A (lower panel). Assays were done using Parkin in a complex with pUb. A colored key for each trace is provided. Coomassie-stained gels of indicated peaks are shown.

The online version of this article includes the following source data and figure supplement(s) for figure 9:

**Source data 1.** Raw image files.

**Figure supplement 1.** The linker connecting REP and RING2 shows conformational flexibility.

**Figure supplement 2.** The linker connecting REP and RING2 domain binds with ubiquitin (Ub$_{don}$) of E2-Ub.

**Figure supplement 2—source data 1.** Raw image files.

the linker region interacting with ubiquitin were highly conserved in Parkin across different species (*Figure 9D*), suggesting their functional importance. Our data in *Figure 2* suggested that RING2 was flexible (open and closed states) mediated by pUbl binding in the basic patch. As R0RBR (R163D/K211N)-pUb complex structure was captured in the closed state of RING2, we wondered whether the linker connecting REP and RING2 may adopt an alternate conformation dependent upon RING2 position (open or closed). The crystallization of the open state of phospho-Parkin remains challenging due to the flexible/multiple possible conformations of the REP-RING2 region. Therefore, we used AlphaFold 2 (*Mirdita et al., 2022*) to predict the model of the linker connecting REP and RING2 of Parkin. Interestingly, the AlphaFold model predicted helical structure in the linker region of Parkin (*Figure 9—figure supplement 1C*) in the RING2 open state of Parkin, indicating the flexible nature of this region under different states (RING2 closed <->RING2 open) of Parkin. The latter also suggested that the conformation of the linker observed in the crystal structure could be one of the intermediates.

To validate the observations from structural analysis, we mutated these residues and compared their ubiquitination activity. In contrast to WT-Parkin, E409A and H422A drastically reduced Parkin activity, whilst I411A, T415N, and K416A resulted in the complete abolishment of Parkin activity (*Figure 9E*). Further inspection revealed that although the linker region of Parkin is not conserved across different members of RBR family E3-ligases (*Figure 9—figure supplement 1D*), hydrophobic nature at the corresponding position of I411 on Parkin is conserved among various RBRs except RNF216 (*Figure 9—figure supplement 1D*). Also, the crystal structures of HOIP, HOIL, HHARI, and RNF216 solved with E2~Ub (*Lechtenberg et al., 2016*; *Horn-Ghetko et al., 2021*; *Wang et al., 2023*) showed interactions between the linker region and donor ubiquitin (Ub$_{don}$) (*Figure 9—figure supplement 2A*). To test whether the linker between REP and RING2 of Parkin binds with donor ubiquitin (Ub$_{don}$), we performed binding assays using E2~Ub$_{don}$. Interestingly, unlike phospho-Parkin, which formed a stable complex with E2~Ub$_{don}$ on SEC and co-eluted with E2~Ub$_{don}$ and phospho-ubiquitin (*Figure 9F*), phospho-Parkin I411A did not show interaction with E2~Ub$_{don}$ (*Figure 9F*). Furthermore, the SEC data was confirmed by ubiquitin-vinyl sulfone (Ub-VS) assay where unlike phospho-Parkin, phospho-Parkin I411A did not react with Ub-VS (*Figure 9—figure supplement 2B*). We also tested Parkin activity using ubiquitin mutants (L71A or L73A), which would perturb the interactions of ubiquitin and Parkin linker as suggested by the structure in *Figure 9C*. Compared to native ubiquitin, ubiquitin mutants showed a loss of Parkin activity (*Figure 9—figure supplement 2C*) which nicely corroborated with our data. Overall, our data showed that the linker region between REP and RING2 interacts with donor ubiquitin and plays a crucial role in Parkin function.

## Discussion

Autoinhibition of Parkin is mediated by several mechanisms. Ubl domain and REP element block the E2 binding site on RING1 (*Chaugule et al., 2011*; *Trempe et al., 2013*; *Kumar et al., 2015*; *Sauvé et al., 2015*), whereas the RING0 domain occludes the catalytic C431 on RING2. A few years after the discovery of Parkin autoinhibition, various groups discovered PINK1-mediated phosphorylation of S65 on the ubiquitin and Ubl domain of Parkin, leading to the activation of Parkin (*Kane et al., 2014*; *Kazlauskaite et al., 2014*; *Koyano et al., 2014*; *Kondapalli et al., 2012*). In the last few years, several

structural studies have aimed to understand the conformational changes in Parkin that are driven by phosphorylation leading to Parkin activation. The structure of RING2 truncated phospho-Parkin (1-382) in complex with pUb showed the pUbl domain of Parkin bound to the basic patch (comprising K161, R163, K211) on RING0, which led to the displacement of RING2 and REP during Parkin activation (*Gladkova et al., 2018*; *Sauvé et al., 2018*). Previous studies using various biophysical methods reported a $K_d$ of ~2 µM between Ubl and R0RBR/ΔUbl-Parkin; however, pUbl showed no interaction, which led to the proposed mechanism suggesting displacement of the pUbl domain to activate Parkin (*Kumar et al., 2015*; *Sauvé et al., 2015*).

Our data show that RING2 and pUbl compete for binding on the basic patch of RING0 (*Figure 2*). Our data also show that RING2 and REP displacement after Parkin phosphorylation is transient; RING2 and REP return to their original position after removal of the pUbl from phospho-Parkin (*Figure 2*). Our data explains that due to the net high concentration of the fused domain (RING2 or pUbl), and competitive mode of interaction, binding/displacement of pUbl/RING2 domain in *trans* couldn't be observed in the previous studies. However, untethering of pUbl/RING2 overcomes the latter issue, and *trans* interaction between Parkin molecules can be observed. By untethering the linker between RING2 and IBR, after pUbl binding, the displaced RING2 is no longer able to return to the RING0 pocket, thus the binding of pUbl on the basic patch of RING0 is stabilized (*Figure 2*). Untethered RING2 leads to a strong affinity between phospho-Ubl and core of Parkin with $K_d$ around 1 µM (*Figures 4 and 5*), which is also supported by complex formation on SEC/SEC-MALS using phospho-Parkin and Parkin (*Figure 5*).

A feedforward control mechanism was suggested in the PINK1-Parkin pathway wherein PINK1-dependent phosphorylation of ubiquitin and Parkin leads to Parkin activation on mitochondria (*Kazlauskaite et al., 2014*; *Shiba-Fukushima et al., 2012*; *Ordureau et al., 2014*; *Zhuang et al., 2016*; *Tang et al., 2017*). However, biophysical studies aimed to understand Parkin activity did not show any dimerization of Parkin or Parkin-Parkin association in *trans* (*Wenzel et al., 2011*; *Walden and Rittinger, 2018*; *Riley et al., 2013*; *Trempe et al., 2013*; *Wauer et al., 2015*; *Kumar et al., 2017*; *Sauvé et al., 2018*; *Condos et al., 2018*; *Wauer and Komander, 2013*; *Gladkova et al., 2018*; *Kumar et al., 2015*; *Sauvé et al., 2015*). Our data demonstrate that phospho-Parkin and WT-Parkin can form a stable complex in *trans* to mediate Parkin dimerization (*Figure 5*). We also show that phospho-Parkin can activate WT-Parkin in *trans*, reaffirming that a major mode of Parkin autoinhibition is mediated by RING0 blocking the catalytic C431 on the RING2 domain. Furthermore, our data suggest an additional feedforward activation model of Parkin wherein fully-activated Parkin (phospho-Parkin bound to pUb) molecules can activate partially-activated Parkin (WT-Parkin bound to pUb) molecules, which is mediated by interactions between pUbl and RING0 in *trans* (*Figure 10*). The latter can be relevant in the context of healthy carriers of heterozygous mutations on Parkin. The critical

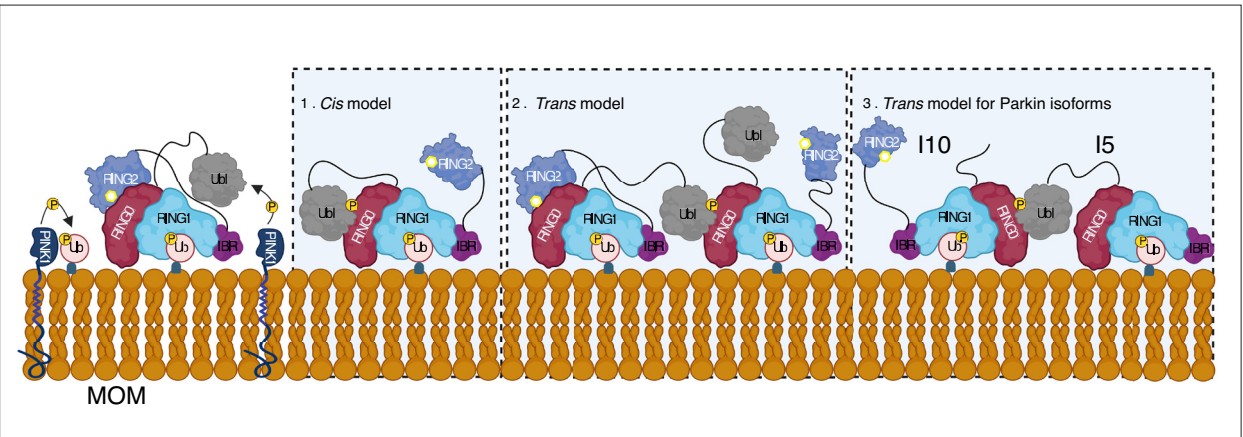

**Figure 10.** Model shows different modes of Parkin activation. The *cis* activation model uses the binding of pUbl in the same molecule, thus resulting in the displacement of RING2 (1). The trans-activation model uses the binding of pUbl of fully activated Parkin (phospho-Parkin complex with pUb) with partially activated Parkin (WT-Parkin and pUb complex), thus resulting in the displacement of RING2 in *trans* (2). Recruitment and activation of Parkin isoforms lacking Ubl (Isoform 10) or RING2 domain (Isoform 5), thus complementing each other using the trans-activation model (3). Catalytic cysteine on RING2 is highlighted.

role of pUbl supports data showing the importance of Ubl phosphorylation in vivo, as demonstrated by the discovery of Parkinson's patients associated with homozygous S65N Parkin mutation (*McWilliams et al., 2018*). This data also highlights the importance of various Parkin isoforms that have been identified (*Figure 5—figure supplement 1*), especially the ones that lack Ubl domain or REP-RING2 domains, as they can complement each other using our proposed *trans* model in *Figure 10*.

ACT was proposed to have a role in Parkin activation, as it was shown that the deletion/mutation of ACT leads to the loss of Parkin activity (*Gladkova et al., 2018*). We demonstrate that ACT plays a key role due to its inherent capacity to bind with the RING0 pocket. Unlike other functional mutations on Parkin affecting interaction with E2 or $Ub_{don}$, ACT deletion does not affect binding with $E2{\sim}Ub_{don}$ (*Figure 7*). We show that ACT plays a crucial role in enzyme kinetics and only slows the Parkin activity, possibly by affecting the inherently dynamic nature of RING2 (*Figure 7*). Furthermore, we also demonstrate that although ACT can be complemented in *trans*, ACT on a *cis* molecule is more effective (*Figure 8*).

The linker connecting IBR and RING2 of Parkin comprises two components: a REP element (391-405) and a flexible linker (408-415). Various Parkin structures solved so far show REP element blocking the E2 binding site on RING1; however, linker (408-415) remained flexible in most structures, and its role remained elusive. Interestingly, pathogenic mutation T415N in the linker region was shown to abolish the E3 ligase activity of Parkin (*Chaugule et al., 2011*). Also, using peptide array analysis, Chaugule and colleagues proposed a Parkin Ubl/ubiquitin-binding (PUB) site in the C-terminal domain of Parkin (*Chaugule et al., 2011*). Here, we demonstrate that the linker (408-415) interacts with donor ubiquitin ($Ub_{don}$) of $E2{\sim}Ub_{don}$ (*Figure 9*). Although the linker between IBR-RING2 is not conserved across RBR family E3-ligases, the core of interactions between the linker and $Ub_{don}$ is mediated by hydrophobic residue in the linker region (*Figure 9*). In the autoinhibited closed state of Parkin, the linker between IBR-RING2 of Parkin is present in a straight conformation, leading to IBR and RING2 occupying diagonally opposite conformation, which is quite similar to what is seen in HOIP RBR and $E2{\sim}Ub_{don}$ complex structure (*Figure 9*, *Figure 9—figure supplement 2A*; *Lechtenberg et al., 2016*). However, the recent structures of RBR family E3-ligases (HHARI, RNF216, HOIL-1) (*Horn-Ghetko et al., 2021*; *Wang et al., 2023*) show a kinked conformation of the linker connecting IBR-RING2 (*Figure 9—figure supplement 2A*). Interestingly, the kink in the linker region plays a crucial role in bringing RING2 to the catalytically feasible state (*Figure 9—figure supplement 2A*). Conversely, under the extended conformation of the linker, catalytic feasibility is not possible (*Figure 9—figure supplement 2A*). The conformational flexibility in the linker (408-415) region of Parkin is also supported by the fact that it is disordered in most Parkin structures, or seen as a loop in a couple of Parkin structures, whereas AlphaFold predicts it as a helix similar to other RBR structures (*Figure 9*, *Figure 9—figure supplement 1*). Previous data observed the opening of RING2 after the addition of $E2{\sim}Ub_{don}$ in R0RBR (*Condos et al., 2018*). The latter observation also suggests that conformational changes might be induced in the linker region after binding with donor ubiquitin or due to the movement of RING2, and needs further investigation. Also, as mentioned above, the conformation of donor ubiquitin and linker captured in the present study might be one of the possible intermediates. Although the regulatory mechanisms vary across RBR family E3-ligases, the catalytic core (IBR-RING2) undergoes similar conformational changes, leading to a unified catalysis mechanism in various RBR family E3-ligases.

Overall, our new structural and biophysical analysis elaborates a new understanding of Parkin activation and regulation that will aid in efforts to develop small molecular activators of Parkin as a therapeutic strategy for PD.

## Materials and methods

**Key resources table**

| Reagent type (species) or resource | Designation | Source or reference | Identifiers | Additional information |
|---|---|---|---|---|
| Strain, strain background (*E. coli*) | DH5α | Invitrogen | Cat.#18265017 | |
| Strain, strain background (*E. coli*) | BL21(DE3) pLysS | Invitrogen | Cat.# C606010 | |
| Cell line | Hela | ATCC | CCL-2 | |

*Continued on next page*

*Continued*

| Reagent type (species) or resource | Designation | Source or reference | Identifiers | Additional information |
|---|---|---|---|---|
| Antibody | Anti-TOMM20 (Rabbit monoclonal) | Abcam | ab186735 RRID:AB_2889972 | IF 1:100 |
| Antibody | Alexa Fluor 405 secondary antibody (Donkey polyclonal) | Thermo Fisher | A-48258 RRID:AB_2890547 | IF 1:1000 |
| Recombinant DNA reagent | pET15b-Parkin (plasmid) | This paper | | See Materials and methods, Molecular biology section |
| Recombinant DNA reagent | pGEX-6P1-Miro1 (plasmid) | This paper | | See Materials and methods, Molecular biology section |
| Recombinant DNA reagent | pET28a-Ph-PINK1 (plasmid) | Addgene | Cat. # 110750 | |
| Recombinant DNA reagent | pET21d-Ube1 (plasmid) | Addgene | Cat. # 34965 | |
| Recombinant DNA reagent | GFP-Parkin plasmid | MRC PPU Reagents & Services | DU23318 | |
| Recombinant DNA reagent | mCherry-Parkin (plasmid) | MRC PPU Reagents & Services | DU77708 | |
| Recombinant DNA reagent | mCherry-Parkin-S65A (plasmid) | MRC PPU Reagents & Services | DU77709 | |
| Recombinant DNA reagent | GFP-Parkin-C431F (plasmid) | MRC PPU Reagents & Services | DU77645 | |
| Recombinant DNA reagent | GFP-Parkin-K211N-C431F (plasmid) | MRC PPU Reagents & Services | DU77659 | |
| Recombinant DNA reagent | GFP-Parkin-H302A-C431F (plasmid) | MRC PPU Reagents & Services | DU77713 | |
| Sequence-based reagent | Hsparkin-TEV-F | This paper | PCR primers | GAGTGCAGTGCCGTATTT GAGAACCTGTATTTTCAG TCACAGGCCTACAGAGTCGAT |
| Sequence-based reagent | Hsparkin-TEV-R | This paper | PCR primers | ATCGACTCTGTAGGCCTGTG ACTGAAAATACAGGTTCTCAA ATACGGCACTGCACTC |
| Sequence-based reagent | Ubl140_pre_F | This paper | PCR primers | AAGTGCTGTTTCAGGGCCC GTCAATCTACAACAGCTTTTATG |
| Sequence-based reagent | Ubl140_pre_R | This paper | PCR primers | CCCTGAAACAGCACTTCCAGT CTACCTGCTGGACTTCC |
| Sequence-based reagent | ParkinK211N_F | This paper | PCR primers | TGCAGAATTTTTCTTTAA TTGTGGAGCACACCC |
| Sequence-based reagent | ParkinK211N_R | This paper | PCR primers | GGGTGTGCTCCACAATT AAAGAAAAATTCTGCA |
| Sequence-based reagent | HsParkinR163D-F | This paper | PCR primers | GTGCAGCCGGGAAAACT CGATGTACAGTGCAGCACCTGC |
| Sequence-based reagent | HsParkinR163D-R | This paper | PCR primers | GCAGGTGCTGCACTGTACATC GAGTTTTCCCGGCTGCAC |
| Sequence-based reagent | Parkin_delACT_F | This paper | PCR primers | GCCCCAGTCAGTCCTCCCA GGAGACTCTGTGGG |
| Sequence-based reagent | Parkin_delACT_R | This paper | PCR primers | GGACTGACTGGGGCTCCC GCTCACAGCCTCC |
| Sequence-based reagent | Parkin_I411A_F | This paper | PCR primers | AAACCGCGAAGAAAACCACCAAGCCCTG |
| Sequence-based reagent | Parkin_I411A_R | This paper | PCR primers | TTCTTCGCGGTTTCT TTGGAGGCTGCTT |
| Sequence-based reagent | Parkin_E409A_F | This paper | PCR primers | CCAAAGCGACCATCA AGAAAACCACCAA |

*Continued on next page*

*Continued*

| Reagent type (species) or resource | Designation | Source or reference | Identifiers | Additional information |
|---|---|---|---|---|
| Sequence-based reagent | Parkin_E409A_F | This paper | PCR primers | ATGGTCGCTTTGG AGGCTGCTTCCCA |
| Sequence-based reagent | Parkin_T415N_R | This paper | PCR primers | AAACCAACAAGC CCTGTCCCCGCT |
| Sequence-based reagent | Parkin_T415N_R | This paper | PCR primers | GGCTTGTTGGTTTTC TTGATGGTTTCTTTG |
| Sequence-based reagent | Parkin_K416A_F | This paper | PCR primers | ACCGCGCCCTG TCCCCGCTGCC |
| Sequence-based reagent | Parkin_K416A_R | This paper | PCR primers | AGGGCGCGGTGGTTT TCTTGATGGTTTCTT |
| Sequence-based reagent | Parkin_H422A_F | This paper | PCR primers | CTGCGCGGTACCAGT GGAAAAAAATGGAG |
| Sequence-based reagent | Parkin_H422A_R | This paper | PCR primers | GTACCGCGCAGC GGGGACAGGGC |
| Sequence-based reagent | Parkin_R | This paper | PCR primers | GGAATTCCTACAC GTCGAACCAGTG |
| Sequence-based reagent | R0RBR_F | This paper | PCR primers | GCGGATCCATCTA CAACAGCTTTTATG |
| Sequence-based reagent | ΔUbl_F | This paper | PCR primers | GCGGATCCGGTCA AGAAATGAATGCA |
| Sequence-based reagent | Miro1_F | This paper | PCR primers | GCGGATCCATGAAA CCAGCTTGTATAAA |
| Sequence-based reagent | Miro1_R | This paper | PCR primers | GCGAATTCTTAAAACG TGGAGCTCTTGAG |
| Commercial kit | Plasmid Extraction Mini Kit | FavorPrep | Cat.# FAPDE300 | |
| Chemical compound | 3-Bromopropylamine hydrobromide | Sigma-Aldrich | Cat.# B79803 | |
| Chemical compound | Vectashield mounting medium | Vector Laboratories | H-1000 | |
| Chemical compound | Carbonyl cyanide 3-chlorophenylhydrazone (CCCP) | Sigma-Aldrhich | C2759 | |
| Chemical compound | DyLight 800 Maleimide | Thermo Fisher Scientific | Cat.# 46621 | |
| Genetic reagent | PEI MAX | Polyscience | 24765–1 | |
| Other | Ni-NTA resin | QIAGEN | Cat.# 30230 | See Materials and methods, Protein purification section |

## Molecular biology

The human *PARK2* gene optimized for bacterial expression of FL-Parkin was cloned in the pET15b vector. Various Parkin mutations used in the present study were made using site-directed mutagenesis (SDM). TEV protease site (ENLYFQS) was substituted in the Parkin construct (between the 382nd-388th residues) as described in *Gladkova et al., 2018*, and an HRV 3C protease site (LEVLFQGP) was inserted (between 140th-141st residues) using site-directed mutagenesis. Ubl (expressing 1-76th amino acids of Parkin) and Ubl-linker (expressing 1-140th amino acids of Parkin) constructs were generated by introducing a stop codon after the 76th and 140th amino acids, respectively, in the FL-Parkin construct. Parkin mutants were generated using site-directed mutagenesis. Miro1 (expressing 181st-592nd) was amplified from the cDNA of the HEK293T cell line using Phusion polymerase (NEB) and cloned into the pGEX-6P1 vector using EcoRI and BamHI restriction enzymes. To generate fluorescently labeled ubiquitin, ubiquitin (residues 2-76) was cloned in a pGEX-6P vector with an overhang expressing GPLCGS at the n-terminal of ubiquitin. For the generation of ubiquitin-3Br protein, the ubiquitin gene (residues 1-75) was cloned in the pTXB-1 vector. Pediculus humanus corporis PINK1 (115 - 575) was a gift from David Komander (*Schubert et al., 2017*) (Addgene plasmid # 110750). Ube1 was a gift from Cynthia Wolberger (*Berndsen and Wolberger, 2011*) (Addgene plasmid # 34965).

## Protein purification

Parkin constructs were expressed in *Escherichia coli* BL21(DE3)pLysS cells. Cells were grown until $OD_{600}$ reached 0.4; the temperature was reduced to 16 °C, and protein was induced by adding 50 μM IPTG, and media was supplemented with 200 μM $ZnCl_2$. Cells were left to grow overnight at 16 °C. Cells were harvested and lysed using sonication in lysis buffer (25 mM Tris pH 7.5, 200 mM NaCl, 5 mM Imidazole, 1 mM β-mercaptoethanol, and 100 μM AEBSF). Protein was purified over Ni-NTA resin. His-Sumo tag was removed using SENP1 protease. Protein was further purified over Hi-Trap Q HP column (GE Healthcare) followed by a gel-filtration column pre-equilibrated with storage buffer (25 mM Tris pH 7.5, 75 mM NaCl, 250 μM TCEP). Other proteins were also purified using similar protocols. PhPINK1 was purified as published before (*Schubert et al., 2017*).

## Isothermal titration calorimetry

Isothermal titration calorimetry (ITC) experiments were performed using PEAQ ITC (Malvern instruments), and data were analyzed using a single-site binding model and competing binding mode. All titrations were performed at 25 °C in 1 X PBS buffer containing 250 μM TCEP. In *Figure 4A*, experiments were done using 350 μM of P-Parkin (K211N) in the syringe and 21 μM of ΔUbl-Parkin in the cell. In *Figure 4B*, experiments were done using 360 μM of P-Parkin K211N in the syringe and 30 μM of untethered ΔUbl-Parkin (TEV)in the cell. In *Figure 5B*, experiments were done using 260 μM of P-Parkin in the syringe and 24 μM of untethered ΔUbl-Parkin (TEV) in the cell.

## Ubiquitination assays

Ubiquitination assays were performed using fluorescently labeled ubiquitin. Ubiquitin labeling was done using Dylight 800 Maleimide (Thermo Scientific), as mentioned previously (*Kumar et al., 2015*), using the manufacturer's specifications. Ubiquitination reactions were performed at 25 °C for 40 min in 25 mM Tris pH 7.5, 50 mM NaCl, 10 mM $MgCl_2$, and 0.1 mM DTT, 10 mM ATP. In all reactions, 25 nM Ube1, 250 nM UbcH7 (E2), 1 μM of E3, and 2 μM of $Ub^{IR800}$ were used in 20 μl of the total reaction volume. 0.5 μM of Ub or pUb was used as an allosteric activator for the experiments in *Figure 3B*, *Figure 9E*, and *Figure 8—figure supplement 1C*. Increasing concentrations of P-Parkin (T270R, C431A; 1 μM, 2 μM, 4 μM, and 8 μM) were used as *trans* activators in *Figure 5E*. The transactivation experiments using pUbl, pUbl-linker, and pUbl-linker-ΔACT were carried out with increasing concentrations of 4 μM, 8 μM, and 16 μM in *Figure 8D, E and F*. Substrate Miro1 ubiquitination reaction was done at 25 °C for 20 min with 5 μM Miro1 and 0.5 μM of E3. Other conditions were the same as mentioned above for ubiquitination/autoubiquitination assay. The reactions were quenched by SDS loading dye and heated at 95 °C for 5 min. The samples were resolved on gradient SDS-PAGE and analyzed using Li-COR Odyssey Infrared Imaging System. Each assay was repeated at least three times. ImageJ software was used to quantify ubiquitination. Bar plots and statistical analysis were done using R.

## Cell culture transfection and microscopy experiment

HeLa cells were cultured in Dulbecco's modified Eagle's medium (DMEM; Gibco) containing 10% (vol/vol) FBS, 1% Pen/Strep, and 1% L-Glutamine at 37 °C under an atmosphere of 5% $CO_2$. Twenty-four-well cell culture plate (35,000 cells/well) was used to seed cells onto borosilicate cover glasses (VWR 631–0148). The following plasmids were generated by MRC Reagent & Services and used to assess Parkin translocation: GFP-Parkin (DU23318), mCherry-Parkin (DU77708), mCherry-Parkin-S65A (DU77709), GFP-Parkin-C431F (DU77645), GFP-Parkin-K211N-C431F (DU77659) and GFP-Parkin-H302A-C431F (DU77713). Transfections were carried out the day after seeding, and plasmids were mixed with PEI (PEI MAX- Polyscience, 24765–1) at a 1:5 ratio in Opti-MEM (Gibco). DNA/PEI mix was left for 45 min at room temperature, then added to the cell cultures and incubated for 48 hr before CCCP treatment (10 μM for 1 hr). For immunostaining, cells were fixed with 4% (wt/vol) paraformaldehyde in PBS for 20 min at room temperature and permeabilized with a blocking buffer containing 3% (wt/vol) Donkey serum and 0.2% (vol/vol) Triton X-100 in PBS for 1 hr. Cells were incubated with TOMM20 (ab186735) primary antibody overnight at 4 °C, followed by incubation with the Alexa Fluor 405 secondary antibody (ThermoFisher, A-48258) for 1 hr at room temperature. After three washes with PBS and a rinse with Milli-Q water, the cover glasses were mounted onto slides using a Vecta-shield mounting medium (Vector Laboratories, H-1000). Microscopy was performed on an LSM 880

laser scanning confocal microscope (ZEISS; Plan-Apochromat 63 x/NA 1.4) using ZEISS Zen Software. Colocalization was assessed using Volocity Software (version 6.3, Quorum Technologies) and determined as Pearson's correlation coefficient for mitochondrial colocalization of GFP and the mitochondrial marker TOMM20. Images were processed using ImageJ software version 1.51 (100).

## Purification of phospho-Ubiquitin (pUb)-3Br

pUb-3Br was purified as published before (*Kumar et al., 2017*; *Borodovsky et al., 2002*). Briefly, ubiquitin (1-75)-Mxe-intein-chitin binding domain was expressed in *Escherichia coli* BL21(DE3) cells using a pTXB-1 vector. Cells were induced at 0.8 O.D. using 250 µM IPTG and incubated at 22 °C for 12 hr. Cells were lysed in lysis buffer (20 mM $Na_2HPO4$ pH 7.2, 200 mM NaCl, 0.1 mM EDTA), and protein was purified using Chitin resin (NEB). The resin was incubated overnight with cleavage buffer (20 mM Na2HPO4 pH 6.0, 200 mM NaCl, 50 mM MESNa, 0.1 mM EDTA) to elute the protein. The eluted protein was reacted with 3-Bromopropylamine hydrobromide (Sigma) at 25 °C for 4 hr. The reacted protein was purified over Hiload 16/600 Superdex 75 pg column (GE Healthcare) pre-equilibrated with 1 X PBS. The fractions containing Ub-3Br were concentrated and phosphorylated using PhPINK1. pUb-3Br was purified over Hiload 16/600 Superdex 75 pg column pre-equilibrated with Parkin storage buffer.

## Synthesis and purification of UbcH7~Ub

The reaction containing 500 µM of UbcH7 (Cys17Ser/Cys86Ser/Cys137Ser), 15 µM of Ube1, and 2.5 mM of 6xHis-Ub in charging buffer (50 mM HEPES pH 7.5, 150 mM NaCl, 10 mM MgCl2, 10 mM ATP) was incubated at 37 °C for 18 hr. The progress of the reaction was monitored over SDS-PAGE. The reaction mixture was passed through Ni-NTA resin to capture His-Ub and UbcH7~Ub (His), and the eluted fraction was purified over Hiload 16/600 Superdex 75 pg column (GE Healthcare). Fractions containing UbcH7~Ub were pooled together and stored for further use.

## Preparation of Parkin complexes for crystallization

In the present study, Parkin complexes with pUb were captured using pUb-3Br. To capture Parkin complexes with pUb-3Br, human Parkin constructs were mutated to include Q347C, as published before (*Kumar et al., 2017*), in various constructs for crystallization experiments. For crystallization of pUbl-linker (1-140) depleted Parkin (141-465) and pUbl-linker (1-140) depleted R0RBR R163D/K211N complex with pUb, Parkin Q347C (3C, TEV) and Parkin R163D/K211N/Q347C (3C) constructs were used, respectively. Proteins were expressed and purified as above. Purified proteins were mixed with pUb-3Br, and Parkin was phosphorylated using PhPINK1 in a phosphorylation buffer containing 5 mM ATP and pUb-3Br. GST-HRV 3C protease was added (in a 1:50 ratio), and proteins were left overnight at 4 °C. The proteins were passed through affinity chromatography to remove GST-HRV 3C protease and PhPINK1. Flow-through was further purified over a gel-filtration column. Fractions containing R0RBR with pUb were pooled together and used for crystallization.

Ternary trans-complex of phospho-Parkin (1–140+141-382 + pUb) was made using Parkin K211N (3C) construct as the donor of pUbl-linker, and R0RBR Q347C (TEV) construct as the acceptor of pUbl-linker. Purified Parkin K211N (3C) was phosphorylated using PhPINK1 as above. Purified R0RBR Q347C (TEV) was treated with His-TEV followed by His-TEV removal over Ni-NTA resin. Twofold molar excess of phospho-Parkin K211N (3C) was mixed with TEV-treated R0RBR Q347C (TEV). The complex containing phospho-Parkin K211N (3C) and R0RB Q347C (141-382) was purified over Hiload 16/600 Superdex 200 pg column pre-equilibrated with Parkin storage buffer. The latter complex was mixed with pUb-3Br and treated with 3C protease. Protein was further purified over Hiload 16/600 Superdex 75 pg column pre-equilibrated with Parkin storage buffer. Fractions containing ternary trans-complex of phospho-Parkin (1–140+141-382 + pUb) were pooled together, concentrated, and used for crystallization.

Ternary trans-complex of phospho-Parkin with *cis* ACT (1–76+77-382 + pUb) was made using the Ubl (1-76) domain of Parkin and ΔUbl-Parkin Q347C (TEV). ΔUbl-Parkin Q347C (TEV) was treated with His-TEV, and His-TEV was removed over Ni-resin. A threefold molar excess of the pUbl domain was mixed with TEV-treated/RING2 untethered ΔUbl-Parkin Q347C (TEV). The pUbl and ΔUbl-Parkin Q347C (77-382) complex was purified over Superdex 75increase 10/300 GL column pre-equilibrated with Parkin storage buffer. The latter trans-complex of phospho-Parkin with *cis* ACT (1–76+77-382)

was mixed with pUb-3Br and purified over Superdex 75 increase 10/300 GL column pre-equilibrated with Parkin storage buffer. Fractions containing ternary trans-complex of phospho-Parkin with *cis* ACT (1–76+77-382 + pUb) were pooled together, concentrated, and used for crystallization.

R0RBR (TEV) was purified as stated above. After treatment with TEV, TEV was depleted using Ni-NTA resin, and untethered R0RBR was purified using Hiload 16/600 Superdex 75 pg column pre-equilibrated with Parkin storage buffer.

## Crystallization and structure determination

Initial crystals of pUbl-linker (1-140) depleted Parkin (141-465) complex with pUb-3Br appeared in 1.6 M Ammonium sulfate, 0.1 M MES monohydrate pH 6.5, and 10% v/v 1,4-Dioxane of HR112 screen (Hampton Research) at 4 °C. Seeding was done to grow good-quality crystals in the same condition. The mother liquor containing 20% (v/v) of glycerol was used as a cryoprotectant for freezing crystals in liquid nitrogen. Crystals of pUbl-linker (1-140) depleted R0RBR (R163D/K211N)-pUb complex appeared in 0.15 M Potassium bromide, and 30% w/v Polyethylene glycol monomethyl ether 2000 of Index screen (Hampton research) at 18 °C. The mother liquor containing 20% (v/v) of PEG 400 was used as a cryoprotectant for freezing crystals in liquid nitrogen. Crystals of ternary trans-complexes of phospho-Parkin were obtained in 0.3 M Sodium nitrate, 0.3 Sodium phosphate dibasic, 0.3 M Ammonium sulfate, 0.1 M Tris (base) & BICINE (pH 8.5), 25% v/v MPD, 25% w/v PEG 1000, and 25% w/v PEG 3350 of Morpheus screen (Molecular dimensions). Good quality crystals were grown at 18 °C using microseeding. The mother liquor containing 10% (v/v) of glycerol was used as a cryoprotectant for freezing crystals in liquid nitrogen. Crystals of untethered R0RBR were grown in 0.1 M HEPES, pH 7.5, 8% PEG 4000, 10% isopropanol, and 0.1 M BaCl2 at 4 °C. The mother liquor containing 20% (v/v) glycerol was used for vitrification.

Data were collected at the European Synchrotron Radiation Facility (ESRF), Grenoble, France. Data were processed using XDS (*Kabsch, 2010*). Scaling was done using Aimless, and the structures were determined by molecular replacement using Phaser, as implemented in CCP-7.1 (*Collaborative Computational Project, Number 4, 1994*). Structures of pUbl-linker (1-140) depleted Parkin (141-465)-pUb complex or pUbl-linker (1-140) depleted R0RBR (R163D/K211N)-pUb complex were solved by using the structure of Pediculus Parkin-phospho-ubiquitin complex (PDBID: 5CAW) as a search model. Structures of ternary trans-complex of phospho-Parkin were solved using phospho-Parkin structure (PDBID: 6GLC) as a search model. Untethered R0RBR structure was determined using R0RBR structure (PDBID: 4I1H) as a search model. The initial model was built and refined using coot (*Emsley and Cowtan, 2004*) and refmac5 (*Murshudov et al., 2011*), respectively.

## Purification of phosphorylated proteins

PhPINK1 was used to phosphorylate various Parkin variants used in the study. Phosphorylation buffer contains 50 mM Tris pH 8.5, 100 mM NaCl, 10 mM MgCl2, 10 mM DTT, and 10 mM ATP. The reactions were performed at 25 °C for 4 hr. Phosphorylation status was checked using Phos-Tag (FUJIFILM) analysis as per the manufacturer's protocol. PINK1 was depleted by affinity chromatography upon completion of the reaction. The phosphorylated proteins were further purified over a gel-filtration column.

## Parkin phosphorylation assay

Parkin phosphorylation assay was performed using 5 µM Parkin and 0.25 µM PINK1 in phosphorylation buffer at 25 °C for 15 min. Increasing concentrations (20 µM, 40 µM, and 80 µM) of pUbl or pUb were added with Parkin to check their effect on Parkin phosphorylation. The samples were analyzed on SDS-PAGE containing Phos-Tag (FUJIFILM) as per the manufacturer's protocol.

## Size-exclusion chromatography

For RING2 or Ubl displacement/binding assays, HRV-3C cleavable and TEV cleavable constructs of Parkin were purified and phosphorylated as above. TEV and HRV 3C were added at the molar ratio (protease: Parkin) of 1:5 and 1:15, respectively. After incubation with respective proteases, proteins were purified using affinity chromatography to remove proteases from Parkin. The proteins were loaded onto Superdex 75 increase 10/300 GL column, and fractions were analyzed over SDS-PAGE.

For the trans-complex assays, phospho-Parkin variants were added in 2-fold molar excess. Also, in all trans-complex assays, the TEV site between IBR and RING2 was present only on the target Parkin

molecules. Furthermore, before complex formation, TEV was removed by affinity chromatography. Proteins were incubated for 30 min at 4 °C before loading onto Superdex 75 increase 10/300 GL column. Fractions were analyzed using SDS-PAGE.

For SEC assay to analyze Parkin interaction with E2~Ub, 10 μM of phospho-Parkin/phospho-Parkin ΔACT/phospho-Parkin I411A was pre-incubated with 15 μM of pUb, followed by the addition of 20 μM of E2~Ub. Proteins were incubated for 1 hr at 4 °C before injecting onto Superdex 75 increase 10/300 GL column. Fractions were analyzed over SDS-PAGE to check the complex formation.

### SEC-MALS

Size-exclusion chromatography (SEC) was performed with inline multi-angle light scattering (MALS) using the Viscotek SEC-MALS 20 system. Protein at 4–6 mg/mL (100 μL) was loaded on P2500-P4000 columns (Malvern) at a flow rate of 0.3 mL/min in buffer containing 20 mM Tris-HCl pH 7.5, 75 mM NaCl, 0.25 mM TCEP. The data were analyzed using OmniSEC 5.11 software.

## Acknowledgements

We thank Prof. Helen Walden for useful discussions. We thank the ESRF, Grenoble, France, and their support staff for providing the beamtime and other logistics support during data collection. We also thank Prof Deepak Nair (RCB, Faridabad) and the Department of Biotechnology (Govt of India) for providing all the logistic support for access to beamtime on ESRF. We thank the central instrument facility (IISER Bhopal) for providing access to the ITC instrument. We acknowledge Mel Wightman (MRC-PPU Reagent & Services) for generating GFP-Parkin and mCherry-Parkin plasmids. We thank the MRC PPU tissue culture team (co-ordinated by Edwin Allen), the MRC PPU Reagents and Services teams (co-ordinated by James Hastie), and the Dundee Imaging Facility (co-ordinated by Paul Appleton). The authors also acknowledge members of the AK group for their feedback on the manuscript and for helping with reagents. DRL is a PMRF fellow. PS is a Senior Research Fellow funded by the Council of Scientific Research and Industrial Research (CSIR). MM was supported by a Wellcome Trust Senior Research Fellowship in Clinical Science (210753/Z/18/Z) and the Michael J Fox Foundation. AK is a recipient of the Innovative Young Biotechnology Award (DBT/12/IYBAl2019/03) and Ramalingaswami Fellowship (DBT/RLF/Re-entry/42/2019), which funded this project. AK also acknowledges IISER, Bhopal, and SERB (SERB/F/6520/2019–2020) for funding.

## Additional information

### Competing interests

Miratul MK Muqit: MM. is a member of the Scientific Advisory Board of Montara Therapeutics Inc and a scientific consultant to MSD UK. The other authors declare that no competing interests exist.

### Funding

| Funder | Grant reference number | Author |
|---|---|---|
| PMRF | 0403018 | Dipti Ranjan Lenka |
| Wellcome Trust Senior Research Fellowship in Clinical Science | 10.35802/210753 | Miratul MK Muqit |
| Michael J Fox Foundation | M.M.K.M. | Miratul MK Muqit |
| Innovative Young Biotechnologist Award | DBT/12/IYBAl2019/03 | Atul Kumar |
| Ramalingaswami Re-entry Fellowship | DBT/RLF/Re-entry/42/2019 | Atul Kumar |
| SERB | SERB/F/6520/2019-2020 | Atul Kumar |
| IISER Bhopal | AK | Atul Kumar |

| Funder | Grant reference number | Author |
|--------|------------------------|--------|

The funders had no role in study design, data collection and interpretation, or the decision to submit the work for publication. For the purpose of Open Access, the authors have applied a CC BY public copyright license to any Author Accepted Manuscript version arising from this submission.

## Author contributions

Dipti Ranjan Lenka, Conceptualization, Resources, Data curation, Formal analysis, Validation, Investigation, Visualization, Methodology, Writing – review and editing, Performed all the biochemical and biophysical experiments, determined crystal structures, and analyzed data; Shakti Virendra Dahe, Investigation, Methodology, Purified various proteins, SEC analysis in Figure 8, and analyzed Parkin isoforms; Odetta Antico, Formal analysis, Investigation, Visualization, Methodology, Writing – review and editing, Performed microscopy experiments and analyzed data; Pritiranjan Sahoo, Methodology, Performed ITC measurements; Alan R Prescott, Methodology, Microscopy data acquisition; Miratul MK Muqit, Formal analysis, Supervision, Funding acquisition, Project administration, Writing – review and editing, Supervised microscopy study; Atul Kumar, Conceptualization, Resources, Data curation, Formal analysis, Supervision, Funding acquisition, Validation, Investigation, Visualization, Methodology, Writing – original draft, Project administration, Writing – review and editing, Refined crystal structures and analyzed data

## Author ORCIDs

Dipti Ranjan Lenka ⓘ https://orcid.org/0009-0009-8890-8331
Shakti Virendra Dahe ⓘ https://orcid.org/0009-0009-8909-240X
Alan R Prescott ⓘ https://orcid.org/0000-0002-0747-7317
Miratul MK Muqit ⓘ https://orcid.org/0000-0001-9733-2404
Atul Kumar ⓘ https://orcid.org/0000-0003-4675-6623

Reviewer #1 (Public Review): https://doi.org/10.7554/eLife.96699.3.sa1
Reviewer #2 (Public Review): https://doi.org/10.7554/eLife.96699.3.sa2
Reviewer #3 (Public Review): https://doi.org/10.7554/eLife.96699.3.sa3
Author response https://doi.org/10.7554/eLife.96699.3.sa4

# Additional files

## Supplementary files
• MDAR checklist

## Data availability

Structure coordinates were deposited in the protein data bank, and accession codes are included in Table 1. All data generated or analyzed in this study are included in the manuscript and supporting files; a manuscript source data file has been provided for all the figures.

The following datasets were generated:

| Author(s) | Year | Dataset title | Dataset URL | Database and Identifier |
|-----------|------|---------------|-------------|-------------------------|
| Lenka et al. | 2024 | Crystal Structure | https://www.rcsb.org/search?request=8IKM | RCSB Protein Data Bank, 8IKM |
| Lenka DR, Kumar A | 2024 | Crystal Structure | https://www.rcsb.org/search?request=8IK6 | RCSB Protein Data Bank, 8IK6 |
| Lenka DR, Kumar A | 2024 | Crystal Structure | https://www.rcsb.org/search?request=8JWV | RCSB Protein Data Bank, 8JWV |
| Lenka DR, Kumar A | 2024 | Crystal Structure | https://www.rcsb.org/search?request=8IKT | RCSB Protein Data Bank, 8IKT |
| Lenka DR, Kumar A | 2024 | Crystal Structure | https://www.rcsb.org/search?request=8IKV | RCSB Protein Data Bank, 8IKV |

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
