## [Editor Report · eLife assessment]

This is a **useful** manuscript describing the competitive binding between Parkin domains to define the importance of dimerization in the mechanism of Parkin regulation and catalytic activity. The evidence supporting the importance of Parkin dimerization for an 'in trans' model of Parkin activity described in this manuscript is **solid**, but lacks more stringent and biochemical characterization of competitive binding that could provide more direct evidence to support the author's conclusions. This work will be of interest to those focused on defining the molecular mechanisms involved in ubiquitin ligase interactions, PINK-Parkin-mediated mitophagy, and mitochondrial organellar quality control.

---

## [Referee Report · Reviewer #1 (Public Review)]

Summary:

The authors used structural and biophysical methods to provide insight into Parkin regulation. The breadth of data supporting their findings was impressive and generally well-orchestrated.

Strengths:

(1) They have done a better job explaining the rationale for their experiments thought-out.

(2) The use of molecular scissors in their construct represents a creative approach to examine inter-domain interactions. Appropriate controls were included.

(3) From my assessment, the experiments are well-conceived and executed.

(4) The authors do a better job of highlighting the question being addressed experimentally.

---

## [Referee Report · Reviewer #2 (Public Review)]

In the revised manuscript, the authors tried to address some of my comments from the previous round of review. Notably, they have performed some additional ITC experiments where protein precipitation is not an issue to probe interactions between PARKIN and different domains. In addition, they have toned down some of the language in the text to better reflect their data and results. However, I still feel that the manuscript lacks some key answers regarding the relative interactions between p-PARKIN and different domains, as discussed in my previous review. A deeper dive into the underlying biophysical and biochemical features that drive these interactions is important to fully understand the importance of their work. However, this manuscript does provide some interesting potential insights into the mechanisms of PARKIN activation that could be useful for the field moving forward.

---

## [Referee Report · Reviewer #3 (Public Review)]

Summary:

In their manuscript, Lenka et al present data that could suggest an "in trans" model of Parkin ubiquitination activity. Parkin is an intensely studied E3 ligase implicated in mitophagy, whereby missense mutations to the PARK2 gene are known to cause autosomal recessive juvenile parkinsonism. From a mechanistic point of view, Parkin is extremely complex. Its activity is tightly controlled by several modes of auto-inhibition that must be released by queues of mitochondrial damage. While the general overview of Parkin activation has been mapped out in recent years, several details have remained murky. In particular, whether Parkin dimerizes as part of its feed-forward signaling mechanism, and whether said dimerization can facilitate ligase activation, has remained unclear. Here, Lenka et al. use various truncation mutants of Parkin in an attempt to understand the likelihood of dimerization (in support of an "in trans" model for catalysis).

Strengths:

The results are bolstered by several distinct approaches including analytical SEC with cleavable Parkin constructs, ITC interaction studies, ubiquitination assays, protein crystallography, and cellular localization studies.

Weaknesses:

As presented, however, the storyline is very confusing to follow and several lines of experimentation felt like distractions from the primary message. Furthermore, many experiments could only indirectly support the author's conclusions, and therefore the final picture of what new features can be firmly added to the model of Parkin activation and function is unclear.

Following peer review and revision, the claims are still not fully supported by direct evidence. While the experimental system may be necessary and/or convenient given the unique challenges in studying Parkin, it does not directly speak toward the conclusions that the authors make, nor does it provide an accurate representation of biology.

---

## [Author Response]

The following is the authors’ response to the original reviews.

**Reviewer #1 (Public Review):**
Summary:The authors used structural and biophysical methods to provide insight into Parkin regulation. The breadth of data supporting their findings was impressive and generally well-orchestrated. Still, the impact of their results builds on recent structural studies and the stated impact is based on these prior works.Strengths:(1) After reading through the paper, the major findings are:- RING2 and pUbl compete for binding to RING0.- Parkin can dimerize.- ACT plays an important role in enzyme kinetics.(2) The use of molecular scissors in their construct represents a creative approach to examining inter-domain interactions.(3) From my assessment, the experiments are well-conceived and executed.

We thank the reviewer for their positive remark and extremely helpful suggestions.

Weaknesses:The manuscript, as written, is NOT for a general audience. Admittedly, I am not an expert on Parkin structure and function, but I had to do a lot of homework to try to understand the underlying rationale and impact. This reflects, I think, that the work generally represents an incremental advance on recent structural findings.To this point, it is hard to understand the impact of this work without more information highlighting the novelty. There are several structures of Parkin in various auto-inhibited states, and it was hard to delineate how this is different.

For the sake of the general audience, we have included all the details of Parkin structures and conformations seen (Extended Fig. 1). The structures in the present study are to validate the biophysical/biochemical experiments, highlighting key findings. For example, we solved the phospho-Parkin (complex with pUb) structure after treatment with 3C protease (Fig. 2C), which washes off the pUbl-linker, as shown in Fig 2B. The structure of the pUbl-linker depleted phospho-Parkin-pUb complex showed that RING2 returned to the closed state (Fig. 2C), which is confirmation of the SEC assay in Fig. 2B. Similarly, the structure of the pUbl-linker depleted phospho-Parkin R163D/K211N-pUb complex (Fig. 3C), was done to validate the SEC data showing displacement of pUbl-linker is independent of pUbl interaction with the basic patch on RING0 (Fig. 3B). In addition, the latter structure also revealed a new donor ubiquitin binding pocket in the linker (connecting REP and RING2) region of Parkin (Fig. 9). Similarly, trans-complex structure of phospho-Parkin (Fig. 4D) was done to validate the biophysical data (Fig. 4A-C, Fig. 5A-D) showing trans-complex between phospho-Parkin and native Parkin. The latter also confirmed that the trans-complex was mediated by interactions between pUbl and the basic patch on RING0 (Fig. 4D). Furthermore, we noticed that the ACT region was disordered in the trans-complex between phospho-Parkin (1-140 + 141-382 + pUb) (Fig. 8A) which had ACT from the trans molecule, indicating ACT might be present in the cis molecule. The latter was validated from the structure of trans-complex between phospho-Parkin with cis ACT (1-76 + 77-382 + pUb) (Fig. 8C), showing the ordered ACT region. The structural finding was further validated by biochemical assays (Fig. 8 D-F, Extended Data Fig. 9C-E).

The structure of TEV-treated R0RBR (TEV) (Extended Data Fig. 4C) was done to ensure that the inclusion of TEV and treatment with TEV protease did not perturb Parkin folding, an important control for our biophysical experiments.

As noted, I appreciated the use of protease sites in the fusion protein construct. It is unclear how the loop region might affect the protein structure and function. The authors worked to demonstrate that this did not introduce artifacts, but the biological context is missing.

We thank the reviewer for appreciating the use of protease sites in the fusion protein construct. Protease sites were used to overcome the competing mode of binding that makes interactions very transient and beyond the detection limit of methods such as ITC or SEC. While these interactions are quite transient in nature, they could still be useful for the activation of various Parkin isoforms that lack either the Ubl domain or RING2 domain (Extended Data Fig. 6, Fig. 10). Also, our Parkin localization assays also suggest an important role of these interactions in the recruitment of Parkin molecules to the damaged mitochondria (Fig. 6).

While it is likely that the binding is competitive between the Ubl and RING2 domains, the data is not quantitative. Is it known whether the folding of the distinct domains is independent? Or are there interactions that alter folding? It seems plausible that conformational rearrangements may invoke an orientation of domains that would be incompatible. The biological context for the importance of this interaction was not clear to me.

This is a great point. In the revised manuscript, we have included quantitative data between phospho-Parkin and untethered ∆Ubl-Parkin (TEV) (Fig. 5B) showing similar interactions using phospho-Parkin K211N and untethered ∆Ubl-Parkin (TEV) (Fig. 4B). Folding of Ubl domain or various combinations of RING domains lacking Ubl seems okay. Also, folding of the RING2 domain on its own appears to be fine. However, human Parkin lacking the RING2 domain seems to have some folding issues, majorly due to exposure of hydrophobic pocket on RING0, also suggested by previous efforts (Gladkova et al., Sauve et al.). The latter could be overcome by co-expression of RING2 lacking Parkin construct with PINK1 (Sauve et al.) as phospho-Ubl binds on the same hydrophobic pocket on RING0 where RING2 binds. A drastic reduction in the melting temperature of phospho-Parkin (Gladkova et al.), very likely due to exposure of hydrophobic surface between RING0 and RING2, correlates with the folding issues of RING0 exposed human Parkin constructs.

From the biological context, the competing nature between phospho-Ubl and RING2 domains could block the non-specific interaction of phosphorylated-ubiquitin-like proteins (phospho-Ub or phospho-NEDD8) with RING0 (Lenka et al.), during Parkin activation.

(5) What is the rationale for mutating Lys211 to Asn? Were other mutations tried? Glu? Ala? Just missing the rationale. I think this may have been identified previously in the field, but not clear what this mutation represents biologically.

Lys211Asn is a Parkinson’s disease mutation; therefore, we decided to use the same mutation for biophysical studies.

I was confused about how the phospho-proteins were generated. After looking through the methods, there appear to be phosphorylation experiments, but it is unclear what the efficiency was for each protein (i.e. what % gets modified). In the text, the authors refer to phospho-Parkin (T270R, C431A), but not clear how these mutations might influence this process. I gather that these are catalytically inactive, but it is unclear to me how this is catalyzing the ubiquitination in the assay.

This is an excellent question. Because different phosphorylation statuses would affect the analysis, we ensured complete phosphorylation status using Phos-Tag SDS-PAGE, as shown below.

**Author response image 1. sa4fig1:** 

Our biophysical experiments in Fig. 5C show that trans complex formation is mediated by interactions between the basic patch (comprising K161, R163, K211) on RING0 and phospho-Ubl domain in trans. These interactions result in the displacement of RING2 (Fig. 5C). Parkin activation is mediated by displacement of RING2 and exposure of catalytic C431 on RING2. While phospho-Parkin T270R/C431A is catalytically dead, the phospho-Ubl domain of phospho-Parkin T270R/C431would bind to the basic patch on RING0 of WT-Parkin resulting in activation of WT-Parkin as shown in Fig. 5E. A schematic figure is shown below to explain the same.

**Author response image 2. sa4fig2:** 

(7) The authors note that "ACT can be complemented in trans; however, it is more efficient in cis", but it is unclear whether both would be important or if the favored interaction is dominant in a biological context.

First, this is an excellent question about the biological context of ACT and needs further exploration. While due to the flexible nature of ACT, it can be complemented both in cis and trans, we can only speculate cis interactions between ACT and RING0 could be more relevant from the biological context as during protein synthesis and folding, ACT would be translated before RING2, and thus ACT would occupy the small hydrophobic patch on RING0 in cis. Unpublished data shows the replacement of the ACT region by Biogen compounds to activate Parkin (https://doi.org/10.21203/rs.3.rs-4119143/v1). The latter finding further suggests the flexibility in this region.

(8) The authors repeatedly note that this study could aid in the development of small-molecule regulators against Parkin to treat PD, but this is a long way off. And it is not clear from their manuscript how this would be achieved. As stated, this is conjecture.

As suggested by this reviewer, we have removed this point in the revised manuscript.

**Reviewer #2 (Public Review):**
This manuscript uses biochemistry and X-ray crystallography to further probe the molecular mechanism of Parkin regulation and activation. Using a construct that incorporates cleavage sites between different Parkin domains to increase the local concentration of specific domains (i.e., molecular scissors), the authors suggest that competitive binding between the p-Ubl and RING2 domains for the RING0 domain regulates Parkin activity. Further, they demonstrate that this competition can occur in trans, with a p-Ubl domain of one Parkin molecule binding the RING0 domain of a second monomer, thus activating the catalytic RING1 domain. In addition, they suggest that the ACT domain can similarly bind and activate Parkin in trans, albeit at a lower efficiency than that observed for p-Ubl. The authors also suggest from crystal structure analysis and some biochemical experiments that the linker region between RING2 and repressor elements interacts with the donor ubiquitin to enhance Parkin activity.Ultimately this manuscript challenges previous work suggesting that the p-Ubl domain does not bind to the Parkin core in the mechanism of Parkin activation. The use of the 'molecular scissors' approach to probe these effects is an interesting approach to probe this type of competitive binding. However, there are issues with the experimental approach manuscript that detract from the overall quality and potential impact of the work.

We thank the reviewer for their positive remark and constructive suggestions.

The competitive binding between p-Ubl and RING2 domains for the Parkin core could have been better defined using biophysical and biochemical approaches that explicitly define the relative affinities that dictate these interactions. A better understanding of these affinities could provide more insight into the relative bindings of these domains, especially as it relates to the in trans interactions.

This is an excellent point regarding the relative affinities of pUbl and RING2 for the Parkin core (lacking Ubl and RING2). While we could purify p-Ubl, we failed to purify human Parkin (lacking RING2 and phospho-Ubl). The latter folding issues were likely due to the exposure of a highly hydrophobic surface on RING0 (as shown below) in the absence of pUbl and RING2 in the R0RB construct. Also, RING2 with an exposed hydrophobic surface would be prone to folding issues, which is not suitable for affinity measurements. A drastic reduction in the melting temperature of phospho-Parkin (Gladkova et al.) also highlights the importance of a hydrophobic surface between RING0 and RING2 on Parkin folding/stability. A separate study would be required to try these Parkin constructs from different species and ensure proper folding before using them for affinity measurements.

**Author response image 3. sa4fig3:** 

I also have concerns about the results of using molecular scissors to 'increase local concentrations' and allow for binding to be observed. These experiments are done primarily using proteolytic cleavage of different domains followed by size exclusion chromatography. ITC experiments suggest that the binding constants for these interactions are in the µM range, although these experiments are problematic as the authors indicate in the text that protein precipitation was observed during these experiments. This type of binding could easily be measured in other assays. My issue relates to the ability of a protein complex (comprising the core and cleaved domains) with a Kd of 1 µM to be maintained in an SEC experiment. The off-rates for these complexes must be exceeding slow, which doesn't really correspond to the low µM binding constants discussed in the text. How do the authors explain this? What is driving the Koff to levels sufficiently slow to prevent dissociation by SEC? Considering that the authors are challenging previous work describing the lack of binding between the p-Ubl domain and the core, these issues should be better resolved in this current manuscript. Further, it's important to have a more detailed understanding of relative affinities when considering the functional implications of this competition in the context of full-length Parkin. Similar comments could be made about the ACT experiments described in the text.

This is a great point. In the revised manuscript, we repeated ITC measurements in a different buffer system, which gave nice ITC data. In the revised manuscript, we have also performed ITC measurements using native phospho-Parkin. Phospho-Parkin and untethered ∆Ubl-Parkin (TEV) (Fig. 5B) show similar affinities as seen between phospho-Parkin K211N and untethered ∆Ubl-Parkin (TEV) (Fig. 4B). However, Kd values were consistent in the range of 1.0 ± 0.4 µM which could not address the reviewer’s point regarding slow off-rate. The crystal structure of the trans-complex of phospho-Parkin shows several hydrophobic and ionic interactions between p-Ubl and Parkin core, suggesting a strong interaction and, thus, justifying the co-elution on SEC. Additionally, ITC measurements between E2-Ub and P-Parkin-pUb show similar affinity (Kd = 0.9 ± 0.2 µM) (Kumar et al., 2015, EMBO J.), and yet they co-elute on SEC (Kumar et al., 2015, EMBO J.).

Ultimately, this work does suggest additional insights into the mechanism of Parkin activation that could contribute to the field. There is a lot of information included in this manuscript, giving it breadth, albeit at the cost of depth for the study of specific interactions. Further, I felt that the authors oversold some of their data in the text, and I'd recommend being a bit more careful when claiming an experiment 'confirms' a specific model. In many cases, there are other models that could explain similar results. For example, in Figure 1C, the authors state that their crystal structure 'confirms' that "RING2 is transiently displaced from the RING0 domain and returns to its original position after washing off the p-Ubl linker". However, it isn't clear to me that RING2 ever dissociated when prepared this way. While there are issues with the work that I feel should be further addressed with additional experiments, there are interesting mechanistic details suggested by this work that could improve our understanding of Parkin activation. However, the full impact of this work won't be fully appreciated until there is a more thorough understanding of the regulation and competitive binding between p-Ubl and RIGN2 to RORB both in cis and in trans.

We thank the reviewer for their positive comment. In the revised manuscript, we have included the reviewer’s suggestion. The conformational changes in phospho-Parkin were established from the SEC assay (Fig. 2A and Fig. 2B), which show displacement/association of phospho-Ubl or RING2 after treatment of phospho-Parkin with 3C and TEV, respectively. For crystallization, we first phosphorylated Parkin, where RING2 is displaced due to phospho-Ubl (as shown in SEC), followed by treatment with 3C protease, which led to pUbl wash-off. The Parkin core separated from phospho-Ubl on SEC was used for crystallization and structure determination in Fig. 2C, where RING2 returned to the RING0 pocket, which confirms SEC data (Fig. 2B).

**Reviewer #3 (Public Review):**
Summary:In their manuscript "Additional feedforward mechanism of Parkin activation via binding of phospho-UBL and RING0 in trans", Lenka et al present data that could suggest an "in trans" model of Parkin ubiquitination activity. Parkin is an intensely studied E3 ligase implicated in mitophagy, whereby missense mutations to the PARK2 gene are known to cause autosomal recessive juvenile parkinsonism. From a mechanistic point of view, Parkin is extremely complex. Its activity is tightly controlled by several modes of auto-inhibition that must be released by queues of mitochondrial damage. While the general overview of Parkin activation has been mapped out in recent years, several details have remained murky. In particular, whether Parkin dimerizes as part of its feed-forward signaling mechanism, and whether said dimerization can facilitate ligase activation, has remained unclear. Here, Lenka et al. use various truncation mutants of Parkin in an attempt to understand the likelihood of dimerization (in support of an "in trans" model for catalysis).Strengths:The results are bolstered by several distinct approaches including analytical SEC with cleavable Parkin constructs, ITC interaction studies, ubiquitination assays, protein crystallography, and cellular localization studies.

We thank the reviewer for their positive remark.

Weaknesses:As presented, however, the storyline is very confusing to follow and several lines of experimentation felt like distractions from the primary message. Furthermore, many experiments could only indirectly support the author's conclusions, and therefore the final picture of what new features can be firmly added to the model of Parkin activation and function is unclear.

We thank the reviewer for their constructive criticism, which has helped us to improve the quality of this manuscript.

Major concerns:(1) This manuscript solves numerous crystal structures of various Parkin components to help support their idea of in trans transfer. The way these structures are presented more resemble models and it is unclear from the figures that these are new complexes solved in this work, and what new insights can be gleaned from them.

The structures in the present study are to validate the biophysical/biochemical experiments highlighting key findings. For example, we solved the phospho-Parkin (complex with pUb) structure after treatment with 3C protease (Fig. 2C), which washes off the pUbl-linker, as shown in Fig. 2B. The structure of pUbl-linker depleted phospho-Parkin-pUb complex showed that RING2 returned to the closed state (Fig. 2C), which is confirmation of the SEC assay in Fig. 2B. Similarly, the structure of the pUbl-linker depleted phospho-Parkin R163D/K211N-pUb complex (Fig. 3C), was done to validate the SEC data showing displacement of pUbl-linker is independent of pUbl interaction with the basic patch on RING0 (Fig. 3B). In addition, the latter structure also revealed a new donor ubiquitin binding pocket in the linker (connecting REP and RING2) region of Parkin (Fig. 9). Similarly, trans-complex structure of phospho-Parkin (Fig. 4D) was done to validate the biophysical data (Fig. 4A-C, Fig. 5A-D) showing trans-complex between phospho-Parkin and native Parkin. The latter also confirmed that the trans-complex was mediated by interactions between pUbl and the basic patch on RING0 (Fig. 4D). Furthermore, we noticed that the ACT region was disordered in the trans-complex between phospho-Parkin (1-140 + 141-382 + pUb) (Fig. 8A) which had ACT from the trans molecule, indicating ACT might be present in the cis molecule. The latter was validated from the structure of trans-complex between phospho-Parkin with cis ACT (1-76 + 77-382 + pUb) (Fig. 8C), showing the ordered ACT region. The structural finding was further validated by biochemical assays (Fig. 8 D-F, Extended Data Fig. 9C-E).

The structure of TEV-treated R0RBR (TEV) (Extended Data Fig. 4C) was done to ensure that the inclusion of TEV and treatment with TEV protease did not perturb Parkin folding, an important control for our biophysical experiments.

(2) There are no experiments that definitively show the in trans activation of Parkin. The binding experiments and size exclusion chromatography are a good start, but the way these experiments are performed, they'd be better suited as support for a stronger experiment showing Parkin dimerization. In addition, the rationale for an in trans activation model is not convincingly explained until the concept of Parkin isoforms is introduced in the Discussion. The authors should consider expanding this concept into other parts of the manuscript.

We thank the reviewer for appreciating the Parkin dimerization. Our biophysical data in Fig. 5C shows that Parkin dimerization is mediated by interactions between phospho-Ubl and RING0 in trans, leading to the displacement of RING2. However, Parkin K211N (on RING0) mutation perturbs interaction with phospho-Parkin and leads to loss of Parkin dimerization and loss of RING2 displacement (Fig. 5C). The interaction between pUbl and K211 pocket on RING0 leads to the displacement of RING2 resulting in Parkin activation as catalytic residue C431 on RING2 is exposed for catalysis. The biophysical experiment is further confirmed by a biochemical experiment where the addition of catalytically in-active phospho-Parkin T270R/C431A activates autoinhibited WT-Parkin in trans using the mechanism as discussed (a schematic representation also shown in Author response image 2).

We thank this reviewer regarding Parkin isoforms. In the revised manuscript, we have included Parkin isoforms in the results section, too.

(2a) For the in trans activation experiment using wt Parkin and pParkin (T270R/C431A) (Figure 3D), there needs to be a large excess of pParkin to stimulate the catalytic activity of wt Parkin. This experiment has low cellular relevance as these point mutations are unlikely to occur together to create this nonfunctional pParkin protein. In the case of pParkin activating wt Parkin (regardless of artificial point mutations inserted to study specifically the in trans activation), if there needs to be much more pParkin around to fully activate wt Parkin, isn't it just more likely that the pParkin would activate in cis?

To test phospho-Parkin as an activator of Parkin in trans, we wanted to use the catalytically inactive version of phospho-Parkin to avoid the background activity of p-Parkin. While it is true that a large excess of pParkin (T270R/C431A) is required to activate WT-Parkin in the in vitro set-up, it is not very surprising as in WT-Parkin, the unphosphorylated Ubl domain would block the E2 binding site on RING1. Also, due to interactions between pParkin (T270R/C431A) molecules, the net concentration of pParkin (T270R/C431A) as an activator would be much lower. However, the Ubl blocking E2 binding site on RING1 won’t be an issue between phospho-Parkin molecules or between Parkin isoforms (lacking Ubl domain or RING2).

(2ai) Another underlying issue with this experiment is that the authors do not consider the possibility that the increased activity observed is a result of increased "substrate" for auto-ubiquitination, as opposed to any role in catalytic activation. Have the authors considered looking at Miro as a substrate in order to control for this?

This is quite an interesting point. However, this will be only possible if Parkin is ubiquitinated in trans, as auto-ubiquitination is possible with active Parkin and not with catalytically dead (phospho-Parkin T270R, C431A) or autoinhibited (WT-Parkin). Also, in the previous version of the manuscript, where we used only phospho-Ubl as an activator of Parkin in trans, we tested Miro1 ubiquitination and auto-ubiquitination, and the results were the same (Author response image 4).

**Author response image 4. sa4fig4:** 

(2b) The authors mention a "higher net concentration" of the "fused domains" with RING0, and use this to justify artificially cleaving the Ubl or RING2 domains from the Parkin core. This fact should be moot. In cells, it is expected there will only be a 1:1 ratio of the Parkin core with the Ubl or RING2 domains. To date, there is no evidence suggesting multiple pUbls or multiple RING2s can bind the RING0 binding site. In fact, the authors here even show that either the RING2 or pUbl needs to be displaced to permit the binding of the other domain. That being said, there would be no "higher net concentration" because there would always be the same molar equivalents of Ubl, RING2, and the Parkin core.

We apologize for the confusion. “Higher net concentration” is with respect to fused domains versus the domain provided in trans. Due to the competing nature of the interactions between pUbl/RING2 and RING0, the interactions are too transient and beyond the detection limit of the biophysical techniques. While the domains are fused (for example, RING0-RING2 in the same polypeptide) in a polypeptide, their effective concentrations are much higher than those (for example, pUbl) provided in trans; thus, biophysical methods fail to detect the interaction. Treatment with protease solves the above issue due to the higher net concentration of the fused domain, and trans interactions can be measured using biophysical techniques. However, the nature of these interactions and conformational changes is very transient, which is also suggested by the data. Therefore, Parkin molecules will never remain associated; rather, Parkin will transiently interact and activate Parkin molecules in trans.

(2c) A larger issue remaining in terms of Parkin activation is the lack of clarity surrounding the role of the linker (77-140); particularly whether its primary role is to tether the Ubl to the cis Parkin molecule versus a role in permitting distal interactions to a trans molecule. The way the authors have conducted the experiments presented in Figure 2 limits the possible interactions that the activated pUbl could have by (a) ablating the binding site in the cis molecule with the K211N mutation; (b) further blocking the binding site in the cis molecule by keeping the RING2 domain intact. These restrictions to the cis parkin molecule effectively force the pUbl to bind in trans. A competition experiment to demonstrate the likelihood of cis or trans activation in direct comparison with each other would provide stronger evidence for trans activation.

This is an excellent point. In the revised manuscript, we have performed experiments using native phospho-Parkin (Revised Figure 5), and the results are consistent with those in Figure 2 ( Revised Figure 4), where we used the K211N mutation.

(3) A major limitation of this study is that the authors interpret structural flexibility from experiments that do not report directly on flexibility. The analytical SEC experiments report on binding affinity and more specifically off-rates. By removing the interdomain linkages, the accompanying on-rate would be drastically impacted, and thus the observations are disconnected from a native scenario. Likewise, observations from protein crystallography can be consistent with flexibility, but certainly should not be directly interpreted in this manner. Rigorous determination of linker and/or domain flexibility would require alternative methods that measure this directly.

We also agree with the reviewer that these methods do not directly capture structural flexibility. Also, rigorous determination of linker flexibility would require alternative methods that measure this directly. However, due to the complex nature of interactions and technical limitations, breaking the interdomain linkages was the best possible way to capture interactions in trans. Interestingly, all previous methods that report cis interactions between pUbl and RING0 also used a similar approach (Gladkova et al., Sauve et al.).

(4) The analysis of the ACT element comes across as incomplete. The authors make a point of a competing interaction with Lys48 of the Ubl domain, but the significance of this is unclear. It is possible that this observation could be an overinterpretation of the crystal structures. Additionally, the rationale for why the ACT element should or shouldn't contribute to in trans activation of different Parkin constructs is not clear. Lastly, the conclusion that this work explains the evolutionary nature of this element in chordates is highly overstated.

We agree with the reviewer that the significance of Lys48 is unclear. We have presented this just as one of the observations from the crystal structure. As the reviewer suggested, we have removed the sentence about the evolutionary nature of this element from the revised manuscript.

(5) The analysis of the REP linker element also seems incomplete. The authors identify contacts to a neighboring pUb molecule in their crystal structure, but the connection between this interface (which could be a crystallization artifact) and their biochemical activity data is not straightforward. The analysis of flexibility within this region using crystallographic and AlphaFold modeling observations is very indirect. The authors also draw parallels with linker regions in other RBR ligases that are involved in recognizing the E2-loaded Ub. Firstly, it is not clear from the text or figures whether the "conserved" hydrophobic within the linker region is involved in these alternative Ub interfaces. And secondly, the authors appear to jump to the conclusion that the Parkin linker region also binds an E2-loaded Ub, even though their original observation from the crystal structure seems inconsistent with this. The entire analysis feels very preliminary and also comes across as tangential to the primary storyline of in trans Parkin activation.

We agree with the reviewer that crystal structure data and biochemical data are not directly linked. In the revised manuscript, we have also highlighted the conserved hydrophobic in the linker region at the ubiquitin interface (Fig. 9C and Extended Data Fig. 11A), which was somehow missed in the original manuscript. We want to add that a very similar analysis and supporting experiments identified donor ubiquitin-binding sites on the IBR and helix connecting RING1-IBR (Kumar et al., Nature Str. and Mol. Biol., 2017), which several other groups later confirmed. In the mentioned study, the Ubl domain of Parkin from the symmetry mate Parkin molecule was identified as a mimic of “donor ubiquitin” on IBR and helix connecting RING1-IBR.

In the present study, a neighboring pUb molecule in the crystal structure is identified as a donor ubiquitin mimic (Fig. 9C) by supporting biophysical/biochemical experiments. First, we show that mutation of I411A in the REP linker of Parkin perturbs Parkin interaction with E2~Ub (donor) (Fig. 9F). Another supporting experiment was performed using a Ubiquitin-VS probe assay, which is independent of E2. Assays using Ubiquitin-VS show that I411A mutation in the REP-RING2 linker perturbs Parkin charging with Ubiquitin-VS (Extended Data Fig. 11 B). Furthermore, the biophysical data showing loss of Parkin interaction with donor ubiquitin is further supported by ubiquitination assays. Mutations in the REP-RING2 linker perturb the Parkin activity (Fig. 9E), confirming biophysical data. This is further confirmed by mutations (L71A or L73A) on ubiquitin (Extended Data Fig. 11C), resulting in loss of Parkin activity. The above experiments nicely establish the role of the REP-RING2 linker in interaction with donor ubiquitin, which is consistent with other RBRs (Extended Data Fig. 11A).

While we agree with the reviewer that this appears tangential to the primary storyline in trans-Parkin activation, we decided to include this data because it could be of interest to the field.

**Recommendations for the authors:**

**Reviewer #1 (Recommendations For The Authors):**
(1) For clarity, a schematic of the domain architecture of Parkin would be helpful at the outset in the main figures. This will help with the introduction to better understand the protein organization. This is lost in the Extended Figure in my opinion.

We thank the reviewer for suggesting this, which we have included in Figure 1 of the revised manuscript.

(2) Related to the competition between the Ubl and RING2 domains, can competition be shown through another method? SPR, ITC, etc? ITC was used in other experiments, but only in the context of mutations (Lys211Asn)? Can this be done with WT sequence?

This is an excellent suggestion. In the revised Figure 5, we have performed ITC experiment using WT Parkin, and the results are consistent with what we observed using Lys211Asn Parkin.

(3) The authors also note that "the AlphaFold model shows a helical structure in the linker region of Parkin (Extended Data Figure 10C), further confirming the flexible nature of this region"... but the secondary structure would not be inherently flexible. This is confusing.

The flexibility is in terms of the conformation of this linker region observed under the open or closed state of Parkin. In the revised manuscript, we have explained this point more clearly.

(4) The manuscript needs extensive revision to improve its readability. Minor grammatical mistakes were prevalent throughout.

We thank the reviewer for pointing out this and we have corrected these in the revised manuscript.

(5) The confocal images are nice, but inset panels may help highlight the regions of interest (ROIs).

This is corrected in the revised manuscript.

(6) Trans is misspelled ("tans") towards the end of the second paragraph on page 16.

This is corrected in the revised manuscript.

(7) The schematics are helpful, but some of the lettering in Figure 2 is very small.

This is corrected in the revised manuscript.

**Reviewer #3 (Recommendations For The Authors):**
(1) A significant portion of the results section refers to the supplement, making the overall readability very difficult.

We accept this issue as a lot of relevant data could not be added to the main figures and thus ended up in the supplement. In the revised manuscript, we have moved some of the supplementary figures to the main figures.

(2) Interpretation of the experiments utilizing many different Parkin constructs and cleavage scenarios (particularly the SEC and crystallography experiments) is extremely difficult. The work would benefit from a layout of the Parkin model system, highlighting cleavage sites, key domain terminology, and mutations used in the study, presented together and early on in the manuscript. Using this to identify a simpler system of referencing Parkin constructs would also be a large improvement.

This is a great suggestion. We have included these points in the revised manuscript, which has improved the readability.

(3) Lines 81-83; the authors say they "demonstrate the conformational changes in Parkin during the activation process", but fail to show any actual conformational changes. Further, much of what is demonstrated in this work (in terms of crystal structures) corroborates existing literature. The authors should use caution not to overstate their original conclusions in light of the large body of work in this area.

We thank the reviewer for pointing out this. We have corrected the above statement in the revised manuscript to indicate that we meant it in the context of trans conformational changes.

(4) Line 446 and 434; there is a discrepancy about which amino acid is present at residue 409. Is this a K408 typo? The authors also present mutational work on K416, but this residue is not shown in the structure panel.

We thank the reviewer for pointing out this. In the revised manuscript, we have corrected these typos.